# The contribution of coral reef-derived dimethyl sulfide to aerosol burden over the Great Barrier Reef: a modelling study

Sonya L. Fiddes[1,2,*], Matthew T. Woodhouse[2], Steve Utembe[3], Robyn Schofield[4], Simon P. Alexander[5], Joel Alroe[6], Scott D. Chambers[7], Zhenyi Chen[8], Luke Cravigan[6], Erin Dunne[2], Ruhi S. Humphries[2], Graham Johnson[6], Melita D. Keywood[2], Todd P. Lane[4], Branka Miljevic[6], Yuko Omori[9,10], Alain Protat[11], Zoran Ristovski[6], Paul Selleck[2], Hilton B. Swan[12], Hiroshi Tanimoto[10], Jason P. Ward[2], and Alastair G. Williams[7]

[1]ARC Centre of Excellence for Climate System Science and the Australian-German Climate and Energy College, University of Melbourne, Australia
[2]Climate Science Centre, Oceans and Atmosphere, Commonwealth Scientific and Industrial Research Organisation, Australia
[3]Environmental Protection Authority Victoria, Australia
[4]ARC Centre of Excellence for Climate Extremes, University of Melbourne, Australia
[5]Australian Antarctic Division, Hobart, Australia
[6]International Laboratory for Air Quality and Health, School of Earth and Atmospheric Sciences, Queensland University of Technology, Australia
[7]Australian Nuclear Science and Technology Organisation, Lucas Heights, New South Wales, Australia
[8]Key Lab of Environmental Optics and Technology, Anhui Institute of Optics and Fine Mechanics, Chinese Academy of Sciences, 230031 Hefei, China
[9]Faculty of Life and Environmental Sciences, University of Tsukuba, Japan
[10]Earth System Division, National Institute for Environmental Studies, Tsukuba, Japan
[11]Australian Bureau of Meteorology, Melbourne, Australia
[12]Faculty of Science and Engineering, Southern Cross University, Australia
[*]Now at the Australian Antarctic Program Partnership, Institute of Marine and Antarctic Studies, University of Tasmania, Australia

**Correspondence:** Sonya Fiddes (sonya.fiddes@utas.edu.au)

**Abstract.** Coral reefs have been found to produce the sulfur compound dimethyl sulfide (DMS), a climatically relevant aerosol precursor predominantly associated with phytoplankton. Until recently, the role of coral reef-derived DMS within the climate system had not been quantified. A study preceding the present work found that DMS produced by corals had negligible long-term climatic forcing at the global-regional scale. However, at sub-daily time scales more typically associated with aerosol and cloud formation, the influence of coral reef-derived DMS on local aerosol radiative effects remains unquantified. The Weather Research and Forecasting - chemistry model (WRF-Chem) has been used in this work to study the role of coral reef-derived DMS at sub-daily time scales for the first time. WRF-Chem was run to coincide with an October 2016 field campaign over the Great Barrier Reef, Queensland, Australia, against which the model was evaluated. After updating and scaling the DMS surface water climatology, the model reproduced DMS and sulfur concentrations well. The inclusion of coral reef-derived DMS resulted in no significant change in sulfate aerosol mass or total aerosol number. Subsequently, no direct or indirect aerosol effects were detected. The results suggest that the co-location of the Great Barrier Reef with significant anthropogenic aerosol

sources along the Queensland coast prevents coral reef derived-aerosol from having a modulating influence on local aerosol burdens in the current climate.

## 1 Introduction

Dimethyl sulfide (DMS) is an important precursor gas for aerosol formation. DMS is produced predominantly by marine organisms such as algae and phytoplankton. Once emitted, atmospheric DMS ($DMS_a$) has a lifetime of approximately 1-2 days (Korhonen et al., 2008; Kloster et al., 2006; Khan et al., 2016) and is primarily oxidised by hydroxyl (OH) and nitrate ($NO_3$) radicals. Oxidation of DMS produces methanesulfonic acid (MSA), dimethyl-sulfoxide (DMSO) and ultimately $SO_2$, which then oxidise further into $H_2SO_4$. $H_2SO_4$ can subsequently condense onto pre-existing particles or, if in sufficiently

high atmospheric concentration and in the absence of pre-existing surfaces, $H_2SO_4$ can nucleate into new particles. In the free troposphere, cooler temperatures, higher supersaturation, and fewer pre-existing particles can provide ideal conditions for aerosol precursor gases to undergo new particle formation. In the boundary layer however, the specific conditions needed for new particle formation occur far less frequently. Merikanto et al. (2009) estimate in the marine boundary layer, only 10% of low level cloud CCN (cloud condensation nuclei) are created in boundary layer nucleation, compared to 45% in the free

troposphere and subsequently entrained to lower levels.

At the global scale DMS plays an important role in global climate modulation via direct (McCormick and Ludwig, 1967) and indirect aerosol effects (Twomey, 1974; Pincus and Baker, 1994; Albrecht, 1989; Warner, 1968). Thomas et al. (2010), Mahajan et al. (2015) and Fiddes et al. (2018) each found global DMS to have a net radiative effect of -2.03 $W\,m^{-2}$, -1.79 $W\,m^{-2}$ and -1.7 $W\,m^{-2}$ at the top of the atmosphere respectively, resulting in surface cooling. Fiddes et al. (2018) further found evidence

of indirect effects predominantly occurring in Southern Hemisphere stratocumulus decks.

Coral reef ecosystems are an unaccounted for source of DMS (Broadbent et al., 2002; Broadbent and Jones, 2004; Jones and Trevena, 2005; Jones et al., 2007; Burdett et al., 2015; Deschaseaux et al., 2019; Jackson et al., 2020b). Many of these studies suggest that the contribution of DMS from coral reefs is a significant source of marine aerosol and may have an impact on local climate. Links to coral reef-derived DMS, aerosol formation, cloud cover and/or sea surface temperatures (SSTs)

have been made by a range of observational studies (Modini et al., 2009; Deschaseaux et al., 2012; Leahy et al., 2013; Swan et al., 2017; Jones et al., 2017; Cropp et al., 2018; Jackson et al., 2018, 2020a). In particular, during a campaign on the Great Barrier Reef (GBR) Modini et al. (2009) describe several instances of boundary layer nucleation events during periods of clean marine air mass. These events resulted in peaks in nucleation mode aerosol number concentrations associated with sulfate and organic aerosol. While acknowledging that the precursor gases could have a continental origin, Modini et al. (2009) suggest

that they are more likely from marine sources. Furthermore, the authors attribute one particularly strong event to precursor gases specifically produced by the GBR. Jackson et al. (2018) show a positive correlation between the satellite derived aerosol optical depth (AOD) and SSTs and irradiance, suggesting the source of sulfur coming from the GBR to be the cause of this result. Similarly, Cropp et al. (2018) also find a positive correlation between satellite derived AOD and coral stress indices taking into account irradiance, tidal activity, and water clarity. Jackson et al. (2020b) further find a significant link of satellite

derived AOD to in-situ $DMS_a$ during periods of calm in daylight hours, suggesting that coral reef production of $DMS_a$ resulted in condensational growth of sulfate aerosol. Furthermore, Jackson et al. (2020b) again report significant correlations of AOD to irradiance and SSTs, citing this as possible evidence for a bioregulatory feedback, which has also been suggested in previous studies (Jones, 2013; Jones et al., 2017; Cropp et al., 2018).

A bioregulatory feedback between coral and DMS would be an example of the CLAW (Charlson, Andreae, Warren and
Lovelock) hypothesis (Charlson et al., 1987). The CLAW hypothesis proposes that marine organisms produce more DMS when stressed, which, when released into the atmosphere undergoes oxidation and participates in direct and indirect aerosol radiative effects by contributing to the formation of sulfate aerosol. Thus by releasing DMS, organisms are able to cool their environment. However, the studies suggesting that coral reefs participate in bioregulatory feedback cannot fully take into account the complexity of the DMS-climate system and its significant non-linearities. Studies that rely on observational products
cannot easily separate the many influences on aerosol loading found in complex coastal regions, for example the influence of sea spray aerosol, continental air masses or intrusions from the free troposphere.

Fiddes et al. (2021) provided the first global modelling study to quantify the climatic influence of coral reefs. This study found that most coral reefs globally were too small to have any effect on aerosol burdens with the exception of those around the Maritime Continent and Australian region. However, although coral reefs in the Australasian region were found to contribute
to nucleation and Aitken mode aerosol, little evidence of a long-term climatic impact could be found over the regions. The results from both Fiddes et al. (2021) and Fiddes et al. (2018) further suggest that regions with large aerosol loading, from either anthropogenic or natural aerosol sources, are unlikely to be sensitive to very small changes in DMS concentrations.

While there have been numerous studies quantifying the global influence of DMS on climate (as discussed in Fiddes et al., 2018), this is not the case for impacts of DMS on regional climate, let alone coral reef-derived DMS. Regional modelling
studies of DMS in the atmosphere and its interactions with local weather and climate are few. DMS has been included in regional climate models (such as the Weather Research and Forecasting model coupled to chemistry, WRF-Chem) since the early 2000s with the incorporation of the Georgia Tech/Goddard Global Ozone Chemistry Aerosol Radiation and Transport (GOCART) scheme (Chin et al., 2000). However DMS is rarely the topic of interest, with little attention being paid to the effects of DMS on aerosol, clouds, and climate. This gap in research is concerning given that DMS has been found in observational
studies to explain a significant part of aerosol variability, especially in clean marine areas (Vallina et al., 2006; Korhonen et al., 2008).

One exception is the major campaign VAMOS (Variability of the American Monsoon Systems) Ocean-Cloud-Atmosphere-Land Study Regional Experiment (VOCALS), which made extensive measurements in the Southeast Pacific stratocumulus deck (off the coast of Chile and Peru). Whilst DMS was not the main focus, Yang et al. (2011) and Saide et al. (2012) evaluate
DMS in WRF-Chem (V3.3) relative to the VOCALS observations. Both these studies suggest that the DMS flux ($flux_{DMS}$) is overestimated in the model when compared to ship based measurements. The overestimation of $flux_{DMS}$ resulted in a high bias in $DMS_a$ concentrations. Whilst these overestimations of $flux_{DMS}$ were considerable, both Yang et al. and Saide et al.'s focus was not on sensitivity testing of DMS, and hence both studies stop short of evaluating how DMS alone may affect aerosol-cloud interaction.

More recently, Muñiz-Unamunzaga et al. (2018) show the importance of including marine halogens and DMS on air quality monitoring for a large coastal city (Los Angeles). The authors note that these inclusions can decrease ozone and nitrogen dioxide levels and can cause large changes in oxidants (OH, $HO_2$ and $NO_3$) and the composition of particulate matter. Studies such as this highlight the importance of DMS not only on clean marine areas, such as those explored in VOCALS, but also in more polluted urban environments. However, Muñiz-Unamunzaga et al. (2018) provide no evaluation of the impact of DMS

itself on the local climate.

In this study, we aim to explore the extent to which coral reef-derived DMS can influence local aerosol burdens over the Great Barrier Reef. We do this by evaluating the ability of WRF-Chem to simulate DMS processes and analysing what the impact of including an additional coral reef source of DMS is on aerosol processes. We evaluate WRF-Chem against new and novel observations from a major field campaign undertaken in the austral spring of 2016: 'GBR as a significant source of

climatically relevant aerosol particles', nicknamed 'Reef to Rainforest' (R2R). The model setup, experiment design and field campaign details are provided in Section 2, while the results of this work are provided in Section 3. We provide a detailed discussion of model limitations in Section 4 and summarise this work in Section 5. Additional methods and evaluation plots can be found in the Appendix.

## 2    Methods

### 2.1    WRF-Chem configuration

WRF-Chem simulations were run for the period: 1st October 2016, at 1200UTC to the 25th October 2016, 1200UTC, to align with the R2R campaign. Two nested domains (one way) were chosen (see Figure 1). The outer domain (D01) covers the majority of the Australian continent and the Coral Sea and is run at a 27 km horizontal grid spacing with a 120 second time step. The inner domain (D02) runs at a 9 km horizontal grid spacing with a 60 second time step and covers the state of

Queensland and the majority of the Great Barrier Reef. All simulations have 41 vertical levels in the troposphere (up to 20.4 km), including 11 levels below 1km, and produce hourly output for each domain.

All WRF-Chem simulations have been meteorologically nudged to provide the best comparison to the R2R field campaign and to ensure that the responses found in the model are attributable to the DMS surface water concentration ($DMS_w$) perturbations and not internal model variability. The Australian Bureau of Meteorology (BoM) Atmospheric high-resolution Regional

Reanalysis for Australia - Regional domain (BARRA-R) has been used to provide initial conditions and to perform nudging at six hourly intervals (Su et al., 2018). Nudging has been applied to temperature and water vapour above the planetary boundary layer and to horizontal wind above vertical level 19 (approximately 3km).

The model was restarted every 4 days (ingesting the previous day's chemical conditions), with a 12 hour spin up thrown out. The chemical boundary and initial conditions are provided by the Model for Ozone and Related Chemical Tracers (MOZART-

4, Emmons et al., 2010). WRF-Chem maps aerosol mass and number to the eight simulated bin sizes (described in Fast et al., 2006) from the bulk aerosol mass provided by MOZART-4, representing the Aitken mode through to the accumulation mode. A full description of the chemistry, aerosol and physics setup for these simulations can be found in Appendix A.

## 2.2 DMS climatologies

The default $DMS_w$ climatology provided by WRF-Chem is the outdated Kettle and Andreae (2000) climatology on a 1x1.25°
grid. The climatology used here is the updated Lana et al. (2011) $DMS_w$ (referred to henceforth as the L11 climatology),
though we note that this has recently been updated again (Hulswar et al., 2021). The interpolation performed by WRF-Chem
(via Prep-Chem) was deemed unsatisfactory (creating unphysical patterns around the coastlines and generally a very coarse
interpolation). For this reason, the L11 climatology for October was interpolated to each WRF-Chem domain using the python
(v3.5) basemap bilinear interpolation, overriding the default WRF-Chem $DMS_w$ climatology. Further smoothing around the
coastlines was performed. All fields that pass through Prep-Chem underwent the same interpolation to a higher resolution for
consistency.

After initial testing, it was found that simulations using L11 overestimated $DMS_w$ in comparison to observations taken
during the R2R campaign (this finding will be described in Section 3). For this reason, a scaled $DMS_w$ climatology was
created, where L11 was divided by 2.8 to match the average $DMS_w$ observations taken during the R2R campaign. The scaled
climatology is referred to as L11S henceforth.

To examine the impact of coral reef-derived DMS, a new source of $DMS_w$ was added to the L11S climatology. The coral reef
source was determined by using the areal fraction of coral reefs per WRF-Chem grid box to add a weighted 50 nM of $DMS_w$,
matching that of the global analysis performed in Fiddes et al. (2021). The 50 nM value was chosen as a high range estimate in
order to maximise any potential signal and response. In reality, this source varies considerably with time (Jones et al., 2007, eg.
up to as much as 54 nM), but is likely much smaller. The coral reef $DMS_w$ source added to the L11S climatology is referred to
as L11SCR and is shown by the coloured contours in Figure 1 along side the R2R $DMS_w$ observations.

## 2.3 Experiment setup

Three simulations are analysed in this study. The first simulation uses the L11 climatology (and hence will be referred to as the
L11 simulation), with no biomass burning or dust and with the Gong et al. (1997) sea salt emissions. While this climatology is
not the default $DMS_w$ climatology in WRF-Chem, it is the most up to date, warranting its evaluation. The second simulation
uses the L11S $DMS_w$ climatology scaled to the ship observations (and subsequently referred to as the L11S simulation). In
addition, L11S uses the Fuentes et al. (2010) sea salt parameterisation which now includes an organic aerosol component that
is excluded under the Gong et al. (1997) parameterisation. Biomass and dust emissions are also included. The third simulation
uses the L11SCR $DMS_w$ climatology and otherwise the same setup as L11S, to examine if coral reef-derived $DMS_w$ plays a
role in aerosol characteristics over the region.

## 2.4 Observations and evaluation methods

The 2016 major field campaign, R2R, aimed to determine if marine aerosol produced by the GBR could affect CCN con-
centrations, cloud formation and subsequently the hydrological cycle, providing essential observational evidence for assessing

DMS-climate interaction. A leading motivation for this field campaign came from observations by Modini et al. (2009). A
selection of the data from this campaign is used in this work to evaluate the WRF-Chem model.

The R2R field campaign took place on two platforms, the first on board the CSIRO Marine National Facility RV Investigator (RVI) which navigated a path around the GBR from the 28th September - 22nd October 2016. The ship track is given in Figure 1. The second platform used in R2R was the Atmospheric Integrated Research Facility for Boundaries and Oxidative Experiments (AIRBOX) mobile atmospheric chemistry lab, stationed at Mission Beach, QLD from the 20th September to the 16th October 2016. The position of AIRBOX is given in Figure 1. While a subset of AIRBOX data has been described previously in Chen et al. (2018) (including lidar, aerosol, trace gas and meteorology data), this is the first work to use the new data set from across the R2R campaign. However, we note that an overview paper on this campaign is currently in preparation (Trounce et al., 2022).

Observations collected both on the RVI and at AIRBOX comprised measurements of meteorology and atmospheric chemical and aerosol composition. These observations aimed to capture each step of the DMS cycle over the GBR for the first time. A list of observations used in this study from the R2R campaign can be found in the Appendix Tables B1 and B2 as well as a brief description of how the data was collected and processed. All observations are available from the relevant institutions upon request.

To compare the estimated flux$_{DMS}$ from the model and observations we used the Liss and Merlivat (1986) parameterisation given modelled and observed DMS$_w$, SSTs and wind fields. To evaluate sulfate aerosol, particle size bins were linearly interpolated to observed particle sizes, assuming aerosol bins are internally mixed. In order to determine periods of time in which the RVI observations were contaminated by ship exhaust, the hourly black carbon concentrations needed to be above $50\,\mathrm{ng\,m^{-3}}$ and the wind direction relative to the ship was $> 120$ and $< 240$ degrees. Any time stamp within $\pm$five minutes of meeting these two criteria were also flagged. Air-masses were considered to have marine origins if radon concentrations were below $300\,\mathrm{mBq\,cm^{-3}}$.

To aid the airmass characterisation, the Hybrid Single Particle Lagrangian Integrated Trajectory Model (HYSPLIT) was used to perform back trajectories (Draxler and Hess, 1997, 1998). The National Centers for Environmental Prediction (NCEP) Global Data Assimilation System (GDAS) 0.5 degree product was used to produce the back trajectories, where vertical motion was calculated using the model vertical velocity. Initial height was set at $100\,\mathrm{m}$ and the trajectories were run for 72 hours, every two hours. Further characterisation of airmasses has been performed by splitting WRF-Chem aerosol into boundary layer and free troposphere masses. The simulated boundary layer height was used. This was done to explore if any specific changes to aerosol could be found in either airmass, in particular due to the differing nucleation processes that occur at the two different levels, and following from Fiddes et al. (2018) who found some impact on nucleation rates in the free troposphere in response to perturbations of coral-reef-derived DMS.

For the WRF-Chem evaluation, time series comparisons, correlations and a bias factor metric have been used, evaluating equivalent model fields to the observations taken during R2R. For this work, evaluation is focused on the WRF-Chem domain two. We note that in both domains, WRF-Chem considers the AIRBOX grid point as ocean (the container was $<50\,\mathrm{m}$ from the shoreline). This has had some impact on how well the model captures the meteorology at AIRBOX (Appendix Figure C2). The

Normalised Mean Bias Factor (NMBF, Yu et al., 2006) is used. The NMBF is a symmetric metric, i.e. negative biases are not
bound by zero, and remains viable when measured values are much smaller than model values. This metric is an improvement
on other model performance metrics, as described by Yu et al. (2006). To ensure clarity of this metric across both positive and
negative biases, the NMBF has been converted into a bias factor (BF) by adding 1 if NMBF $< 0$, or subtracting 1 if NMBF
$> 1$. Correlation analysis has been performed using Spearman's rank correlation methods (Wilks, 2011). This method is a
non-parametric test that quantifies the monotonicity of the relationship between two variables.

## 3  Results

### 3.1  Surface water DMS and the resulting sea-air flux

Figure 2 shows the timeseries of $DMS_w$, wind speed and the resultant $flux_{DMS}$ for the three WRF-Chem simulations and the
RVI observations. In Figure 2a, the effect of scaling the L11 $DMS_w$ climatology can clearly be seen, where L11S represents
a more realistic value for the GBR region with a BF of 1.02 compared to L11 of 2.74. While the L11 climatology is, as of
writing, the most up to date gridded data set of $DMS_w$ available we can see that for this region it significantly overestimates
DMS. This result highlights that much greater sampling of $DMS_w$ variability over space and time is required, especially in
regional studies. We note a new $DMS_w$ climatology has just been submitted for review Hulswar et al. (2021) and that Lana
et al. (2011) is not the default climatology for WRF-Chem, rather the original Kettle et al. (1999) climatology.

Figure 2b shows that overall the model predicts wind speed along the RVI path well, with L11S having a NMBF of 1.06, and
a correlation coefficient of R=0.87 to the observations. The diurnal cycle is also well captured (not shown) although the model
tends to underestimate and delay the peak wind speed. This is attributed to the poor simulation of the sea breeze structure
(results not shown). Some model skill may be attributed to the fact that RVI observations are assimilated into the BARRA
reanalysis product used to nudge the model, although we also note that model winds are free running below 3 km. Comparison
of wind roses (not shown) indicates that the model has a bias of winds from the south-east. This bias reflects the predominant
large scale flow over the area for this time of year. Generally however, the model captures the overall meteorology as well as
can be expected for the RVI (see Appendix Figure C1), though less satisfactorily for AIRBOX (Appendix Figure C2).

Wind speed is a key factor in the Liss and Merlivat (1986) flux parameterisation. An estimate of the $flux_{DMS}$ along the RVI
track is shown in Figure 2c. Here we can see that despite the model's constant $DMS_w$, it is able to do a comparatively good
job representing the average $flux_{DMS}$, with a BF of 1.21 for L11S, compared to 2.32 for L11. This skill is predominantly due
to the well captured marine wind speeds discussed above. Visually, the L11S $flux_{DMS}$ timeseries appears to follow that of the
observations relatively closely, although with a weak R value of 0.24 and underestimated variance (where $\sigma_{L11S}$ = 2.77, $\sigma_{Obs}$
= 3.11), likely due to the constant $DMS_w$.

From Figure 2a, we can see where the ship is within a grid box that has an additional coral reef $DMS_w$ source included in
L11SCR. The corresponding $flux_{DMS}$ is in general much larger than what was observed although we recognise that the ship
did not measure directly over coral reefs. Nevertheless, this is a clear demonstration of how additional coral reef $DMS_w$ is
influencing the $flux_{DMS}$ and should subsequently influence $DMS_a$.

## 3.2 Atmospheric DMS and sulfate aerosol mass

Figure 3a and b shows $DMS_a$ from the RVI and at AIRBOX. The average at both sites is moderately well captured. For the RVI, the BF for L11S compared to the observations is 1.29, while at AIRBOX this is 1.42. Weak (negative) and insignificant correlations between the observations and simulations are found for both locations. The poorer performance of the model at AIRBOX may be due to the complexity of the location on the coast. However, the weak and negative correlations suggest that the model is missing an important aspect of $DMS_a$ variability, likely caused by the constant $DMS_w$ field or perhaps missing chemical sinks. While WRF-Chem has more complex DMS chemistry compared to other chemical models, comprising 30 DMS oxidation pathway reactions, it is possible that it is still missing important reactions. For example, the importance of DMS removal by BrO or $Cl_2$ has been highlighted in the literature (Breider et al., 2010; Khan et al., 2016; Muñiz-Unamunzaga et al., 2018). Specifically, Khan et al. (2016) note that without these inclusions, the variability of $DMS_a$ is not well modelled.

Nevertheless, significant improvement can be seen in $DMS_a$ in the L11S simulations compared to the standard L11 simulation. The addition of coral reef $DMS_w$ has made a small difference in $DMS_a$ when near reef regions. However, the significant changes to $DMS_w$ between L11, L11S and L11SCR have not had a significant impact on the sulfate aerosol mass, shown in Figure 3c and d.

For L11S, over the entire timeseries, the model underestimates the observations of sulfate aerosol mass by approximately $0.11\,\mu g\,m^{-3}$, with a BF of 1.3. A moderate correlation of R = 0.46 suggests that the model is capturing some of the observed trends and variability. For AIRBOX, the underestimation of the observations by L11 increases to $0.16\,\mu g\,m^{-3}$ and the BF is 1.43. In addition, a statistically significant negative relationship is found between the two timeseries (R=-0.16). This is likely due to a number of local sulfate sources observed by AIRBOX that were not included in the model, for example, vehicle emissions, that impact the variability of the timeseries.

Importantly, we note that reducing $DMS_w$ by approximately 65% between L11 and L11S results in a decrease of only 10% in total surface sulfate aerosol mass along the RVI track. It is possible that this is because L11S also included biomass burning, while L11 did not. The mean difference of $SO_4^{2-}$ surface concentration for L11 and L11S over the entire RVI timeseries is $-0.039\,\mu g\,m^{-3}$. When periods of contaminated air (airmasses that contained high levels of black carbon, terrestrial influence or were flagged for other reasons) are filtered out of the calculations, the difference between L11 and L11S was $-0.049\,\mu g\,m^{-3}$, or a -15% change. The difference between the filtered and unfiltered means suggests that the inclusion of biomass burning has offset the sulfate reduction caused by $DMS_w$ only marginally (by about 5%). Nevertheless, these results imply that DMS only plays a small role in the sulfate aerosol burden over the GBR.

## 3.3 Terrestrial airmass influence

Figure 4a shows the time series of radon and black carbon (for both the RVI and L11S), the mean wind direction in the model and three sets of HYSPLIT trajectories over Stations 3.1, 3.2 and 4. It is clear from the radon timeseries that over the campaign, the ship did not encounter what could be considered clean marine air often (defined as periods below $300\,mBq\,m^{-3}$), although we note that exposed coral reef atolls are also a source of radon. The radon time series is coloured by the exhaust contamination

flag and indicates that there were even fewer occasions in which conditions uninfluenced by ship exhaust (shown by the green colours) or terrestrial air mass were measured.

Comparing the RVI black carbon to the L11S concentrations, where ship exhaust from the RVI is not included, we can see some agreement in periods of terrestrial airmass (eg. between stations 3.1 and 3.2). While the L11S black carbon levels are lower than what was measured, the mean (in periods of no exhaust contamination) of $0.07\,\mu\mathrm{g\,m^{-3}}$ over the time period shown is above that of which was considered clean in this study ($0.05\,\mu\mathrm{g\,m^{-3}}$) also implying a predominant terrestrial influence. We note that Appendix Figure C4 shows an evaluation of BC at AIRBOX, where modelled BC is also lower than that of the observations (by a factor of approximately 3.3). The majority of this underestimation is due to some very large peaks in the observations which Chen et al. (2018) attributed in part to biomass burning, but may also be a result of local vehicle movements. This may indicate that the model is not capturing some small-scale or transient terrestrial/anthropogenic emissions. This limitation is not detrimental to the results.

In Figure 4b, south easterly surface winds are shown to prevail, although some bias in the day to day wind direction was found compared to the RVI (see Appendix Figure C1). Looking at this map, one may expect that this period did comprise clean marine airmasses, and, as an example, this time of year was in part chosen due to this prevailing wind direction. However, as the 72 hour HYSPLIT back trajectories in Figure 4c-e show, despite the wind coming from the south east, much of the time, this airmass had actually recirculated over the Australian continent and spent only a partial time over the ocean. The dominance of terrestrial airmass during the R2R campaign was also demonstrated by Chen et al. (2019), where much of the airmass measured at AIRBOX had strong terrestrial influence, with signatures of biomass burning.

In addition to unfavourable synoptic conditions, one such cause of a lack of clean marine periods is the influence of sea-land breeze coupling. Over coastal-marine regions recycling of airmasses over the land and ocean can occur and have far reaching impacts. For example, a sea-land breeze circulation up to $150\,\mathrm{km}$ offshore under favourable conditions during the R2R campaign was detected in previous (unpublished) WRF simulations (personal communication C. Vincent). Similarly, sea-land coupling was also observed at AIRBOX (not shown). This circulation between land and ocean can lead to terrestrial airmasses extending far offshore.

## 3.4   Dominant anthropogenic sources of sulfur

Now we consider not just how DMS has changed in response to the simulated coral reef perturbations over the RVI track or at AIRBOX, but how it has changed over the entire WRF domain, and in particular over coral reef regions. To be clear - the changes in DMS are simulated only - not observed changes. A vertical transect of $\mathrm{DMS}_a$ and the surface mean for the entire domain is shown in Figure 5a and b. In this figure, the source of coral reef $\mathrm{DMS}_a$ is clearly evident in the (simulated) boundary layer (c) and being blown with the prevailing model winds at the surface in (d). Over the entire domain, at the surface, a significant increase in $\mathrm{DMS}_a$ of $0.003\,\mathrm{ppb}$ is found, approximately 12%, with a mean increase of 40% found over coral reef grid points. These significant increases in $\mathrm{DMS}_a$ however, do not result in significant change in sulfate aerosol mass, as found along the RVI track and at AIRBOX.

We note that the model captures the boundary layer height along the RVI track relatively well (Appendix Figure C1), with some exceptions due in part to the model and in part to the physical limitation of the lidar system resulting in no boundary layer height detection below 500m altitude as well as possible miss-identification of some cloud layers by the lidar algorithm (eg - spikes in boundary layer height in the early time-series). At AIRBOX, the boundary layer height is not as well captured (Appendix Figure C2), in agreement with Chen et al. (2019). This result also agrees with previous work that indicates the Mellor-Yamada-Janjic (MYJ) boundary layer scheme underestimates the boundary layer height the most in coastal marine areas, which then improves further offshore (Rahn and Garreaud, 2010).

Figures 6a and b show the vertical transect and surface mean of total sulfate aerosol mass (including in-cloud sulfate aerosol). These two plots clearly demonstrate that anthropogenic emissions represent a significant source of sulfate aerosol (among other species) for the GBR region. In Figure 6a, over the Gladstone coal fire power station (brown triangle), a dominant plume in sulfate aerosol can be seen, while in Figure 6b, numerous plumes, that align with known power generators, can be seen. We note that the same delineation between the boundary layer and free troposphere found in the $DMS_a$ plots is not seen in the sulfate aerosol mass.

Figure, 6c and d indicate no coherent change in total sulfate aerosol mass that could be robustly attributed to the inclusion of coral reef-derived DMS. The surface mean change between L11S and L11SCR over the entire domain is $0.0018\,\mu g\,m^{-3}$, or a change of 0.38%. Directly over coral reef grid points, an increase of 0.47% was found.

### 3.5 Nucleation pathways of coral reef-derived DMS

The prevalence of anthropogenic sulfur and the abundance of pre-existing particles suggests that the small addition of sulfate from DMS is unlikely to participate in new particle formation in the boundary layer. Rather, it is more likely that coral reef-derived $H_2SO_4$ would condense onto pre-existing particles, growing their mass. Below, we analyse column integrals of aerosol number and mass in the free troposphere and boundary layer to test this hypothesis. We have separated the two atmospheric profiles as Fiddes et al. (2021) noted larger changes in the free troposphere in response to perturbed coral reef DMS than in the boundary layer.

Figure 7 shows the transect of the boundary layer (left) and free troposphere (right) total column sulfate mass and aerosol number for the bins representing Aitken and accumulation mode aerosol. We suggest that regions where the changes along the transects in the number and mass co-vary are likely due to internal model variability (eg. some variation between model simulations independent of perturbations, despite the nudging, is expected), rather than changes in the $DMS_a$ field. In the larger bins (Figure 7e-j), this appears to be the case in most locations. However, in the smaller bins (Figure 7a-d), some instances where the mass has changed independently of the number are found.

In Figure 7a and c, the total boundary layer total column sulfate mass has increased in some areas, while the number has not. These increases may be evidence of condensational growth. Importantly we note that the regions over which the increase in mass occur are not co-located with coral reefs, but may be due to advected $DMS_a$ and its oxidation products.

In the free troposphere, new particle formation is far more likely to occur. Examining the smallest bin size (Figure 7b) however, no clear evidence of the coral reef-derived sulfuric acid participating in new particle formation is found. Rather,

directly over coral reefs, a relatively large decrease in sulfate aerosol mass is found compared to the simulation without coral-reef-derived DMS, with a lesser reduction in particle number concentration. These decreases may suggest that the coral reef-derived precursors have grown existing small sized particles, causing them to shift into larger bin sizes. On average over the transects for the remaining bin sizes, increases in free tropospheric sulfate mass was found, accompanied by decreases in number (keeping in mind that these changes are insignificant and less than 1%). This may further suggest greater coagulation rates, reducing the number, while increasing the mass. However, due to the very small and insignificant changes found, we have low confidence that these results are caused directly by coral reef-derived DMS as opposed to model noise.

While locally, changes in sulfate mass in some cases appear to be up to 5%, on average the changes over the transects in either the boundary layer or free troposphere are less than 1%. Furthermore, the largest signals from coral reef-derived DMS appear to occur in the smallest size aerosol, with little discernible change in the larger aerosol sizes that are of greater climatic relevance. Examination of changes in cloud condensation nuclei (CCN, not shown) confirm this and indicate that the very small addition of sulfate by corals is unlikely to have any direct or indirect aerosol effects over the GBR region. Further investigation of these aerosol effects has been carried out (not shown), and no significant changes in clear sky radiation, total radiation, cloud droplet number, liquid water path, cloud fraction or precipitation were found.

## 4    Discussion

An important limitation of this present study is the choice of model setup in relation to the aerosol size distribution. In this work, the default aerosol size bins were used, which range from 39 nm to 10 $\mu$m (Fast et al., 2006), capturing aerosol in the Aitken mode through to the coarse mode. This set-up does not explicitly represent nucleation mode aerosol, which is instead parameterised to contribute directly to Aitken mode via growth. The parameterisation of nucleation mode particles will be discussed in greater detail below and is described in Appendix A. A model set-up in which the nucleation mode was explicitly resolved was not used in the present study as we followed the model set-up of Yang et al. (2011) and Saide et al. (2012), one of the few studies in the literature using WRF-Chem to examine the role of DMS in climate.

The impact of the nucleation mode configuration on the results of this study could be two fold. Firstly, Lee et al. (2013) demonstrated that by not explicitly including nucleation and early growth (including coagulation) of nucleation mode particles can significantly overestimate the N10 (number of particles with a dry diameter greater than 10 nm). However, only minor impacts on CCN and the subsequent indirect aerosol effects were found when nucleation mode processes below 10 nm were explicitly resolved. Such results have not been discussed in detail in this study, but we can briefly confirm that total aerosol number concentrations (N10 and CCN) were over-estimated for this work (see Appendix Figures C3 and C4).

In our previous work (Fiddes et al., 2021), the GLObal Model of Aerosol Processes - mode (GLOMAP) aerosol scheme was used, which explicitly included nucleation mode aerosol below 5 nm. New particular formation was parameterised via the Kulmala et al. (1998) scheme in the free troposphere and organic mediated (Metzger et al., 2010) boundary layer nucleation scheme. In that work, a small impact on nucleation-Aitken mode aerosol was found when coral-reef-derived aerosol were included, but these increases were not large enough to conclusively be found to have altered the CCN. It was also found that

the largest differences in particles with a dry diameter greater than 3 nm was were in the free troposphere. This present work, using the same coral-reef-derived DMS contributions, also found a small change in Aitken mode aerosol, and little to no change in CCN. A fully resolved nucleation mode may result in fewer 'ultra-fine' (below 70 nm) aerosol and change how coral-reef-derived aerosol may interact with the ambient aerosol burden. However, in light of our previous work, we do not have reason to believe that the conclusions of this work would be significantly altered for the present day climate.

Secondly, a number of studies have reported the impact of including new particle formation on clouds and radiation, where new particle formation increases the condensation sink (Blichner et al., 2021) and reduces the amount of sulfuric acid available for condensation (Sullivan et al., 2018), resulting in inhibited growth of particles into CCN, thereby reducing the cloud albedo. This mechanism is further supported by Gordon et al. (2016), where, under pre-industrial conditions, the inclusion of biogenic new particle formation results in large decreases in cloud albedo. Blichner et al. (2021) also find that under present-day con-
ditions, new particle formation, including early growth, has a greater influence on the activation of cloud droplets than under pre-industrial conditions in part due to the increased sulfate burdens and associated increased hygroscopicity, resulting in CCN activation at smaller sizes.

In this work, no discernible effect was found on cloud or radiative properties. Whether this is caused by the model setup or is because the coral-reef-derived source of aerosol is insignificant is difficult to conclude in this study alone. However, Fiddes
et al. (2021) also found little-to-no effect on cloud-radiative properties, indicating that the results here are robust.

In addition to the limited aerosol bin sizes discussed above, the choice of nucleation mechanism, whether parameterised or more fully resolved, is likely of some importance to this work. Much community effort has been undertaken on improving our understanding of new particle formation, including new observations as well as updated parameterisations (Lee et al., 2019; Semeniuk and Dastoor, 2018). Here, the Wexler et al. (1994) parameterisation is used, in which only binary nucleation of
sulfuric acid with water is included. Current understanding suggests that binary nucleation cannot satisfactorily parameterise sulfate concentrations in the clean marine boundary layer (Semeniuk and Dastoor, 2018). To add to this, ternary nucleation processes (with ammonia) have been found to under predict new particle formation (Semeniuk and Dastoor, 2018; Zaveri et al., 2008).

These findings demonstrate how complex new particle formation is and how much we still need to understand. For example,
literature suggests that nucleation involving amines, ammonia, iodine, or other ions with DMS or MSA may be important for coastal regions such as Mace Head or the Antarctic Peninsula (Brean et al., 2021; O'Dowd et al., 2002; Semeniuk and Dastoor, 2018). Other work has suggested that the presence of isoprene, a compound emitted in large quantities by eucalypt forests (Emmerson et al., 2020), may suppress new particle formation (Lee et al., 2019). How important these compounds and interactions are for the GBR, a tropical coastal marine region bounded by significant eucalypt forests, has not been quantified.
While these processes are not included in our work, we hope that this present study can be used as a beginning point to understand the aerosol processes of the region more fully.

In summary, a more representative aerosol scheme is desirable, in particular to better understand the possibility/influence of boundary layer nucleation, as suggested to be possible by Modini et al. (2009). Though the set-up of the current work may have precluded the ability of boundary layer nucleation to realistically occur, our previous work suggests that nucleation in

the free troposphere is more important, and hence we do not expect significantly different results. Never-the-less, we strongly recommend any future work similar to the present study employs a more fully resolved size distribution similar to Matsui et al. (2011), Zhao et al. (2020) or Sullivan et al. (2018) within WRF-Chem or using alternate regional models such as in Gordon et al. (2018), and if available, a nucleation mechanism that considers the complexity of a coastal marine environment.

  Finally, further limitations of this study include the nudged meteorology (every six hours), potentially limiting the ability of
indirect aerosol effects to occur and any possible feedbacks. As shown in Fiddes et al. (2021), large differences can be found in aerosol-climate processes between nudged and free running simulations, although differentiating between model noise and a real signal is difficult. However, we do not expect that in a similar free running study the results would be significantly changed due to the very small changes in aerosol found in this work, the abundance of anthropogenic aerosol, and the fact that the meteorological nudging was not applied in the boundary layer. Furthermore, the four day restarts of WRF-Chem, despite
ingesting the previous day's chemistry, appeared to impact total aerosol numbers, including CCN, which were found to be strongly constrained by this set-up choice. This impact has limited the analysis of these fields in this study. The restarts were not thought to impact the DMS processes, in part due to the lifetime of DMS in the atmosphere.

## 5 Conclusions

  Coral reefs as an unaccounted-for source of DMS have gained attention over recent years, with numerous observational studies
suggesting they play an important and even regulatory role in local climate (Jones, 2013; Hopkins et al., 2016; Jones et al., 2017; Cropp et al., 2018; Jackson et al., 2018, 2020b). While Fiddes et al. (2021) in a global modelling study found that coral reef derived DMS over the Maritime Continent and Australian region had little impact on long-term climate, no regional-scale modelling has been performed prior to this present study. This is particularly important if we are considering temporal and spatial resolutions relevant to bioregulatory feedbacks. In this work, we have evaluated the ability of WRF-Chem to simulate
DMS and sulfur processes and tested the sensitivity of these processes to perturbations in $DMS_w$, with a particular focus on coral reef-derived DMS.

  We find that, compared to observations taken during the R2R campaign, the Lana et al. (2011) climatology significantly overestimates $DMS_w$ and required reduction by 65% to be of a similar magnitude. This finding adds to a growing argument of a need for an updated and if possible, time-varying (beyond the fixed monthly climatology) $DMS_w$ climatology (e.g. as
suggested in Green and Hatton, 2014). We note that a third generation climatology has recently been developed (Hulswar et al., 2021), however this still does not address the need for time-varying data sets. Furthermore, our finding demonstrates that greater attention needs to be paid to the $DMS_w$ climatology within modelling systems, highlighted by the fact that the default WRF-chem climatology is the out of date Kettle and Andreae (2000) climatology.

  With a $DMS_w$ field that aligns with observations, the Liss and Merlivat (1986) $flux_{DMS}$ calculated from both observations
and the model agree reasonably well in magnitude. This result is in part due to the well-captured marine wind speeds along the RVI track. Subsequently, $DMS_a$ is also reasonably well captured, although overestimated, as is sulfate aerosol mass over the ship track. The Liss and Merlivat (1986) flux parameterisation is considered a conservative parameterisation compared to

other methods which provide much larger fluxes (see Appendix for details). Hence the overestimation of $DMS_a$ found here (despite matched $DMS_w$ and well captured wind speeds) further suggests that Liss and Merlivat (1986) is the most realistic parameterisation for calculating the flux$_{DMS}$. Nevertheless, this evaluation gives us confidence that the model is able to capture the key processes in the DMS-aerosol system.

By comparing simulations with the original Lana et al. (2011) $DMS_w$ climatology to the scaled climatology, we find that DMS plays only a small role in sulfate aerosol burdens over the GBR. For a 65% reduction in $DMS_w$, a subsequent 67% reduction in $DMS_a$ and a 10-15% reduction in sulfate aerosol mass was found at the surface. Examination of the background meteorological conditions indicate that influence from terrestrial airmasses occurred for the majority of the R2R campaign, which was broadly reflected in the WRF-Chem model. Furthermore, we suggest that local anthropogenic sources of sulfur from fossil fuel power generation is likely to have a strong influence over the GBR airmass due to proximity, interaction of the sea breeze and synoptic conditions. We recommend further observational studies are carried out to confirm if this is the case for different times of year. Additionally, we note that major coral bleaching events occurred in the summer prior to this field campaign. While the coral reefs south of Cairns (the region of this campaign) were not as severely bleached, we cannot rule out an impact on the production of DMS by coral reefs due to this event.

These results suggest that much smaller changes in DMS from coral reefs are unlikely to have a large impact on the aerosol burden. We find that by adding in a source of coral reef DMS, the total sulfate aerosol mass increases by less than 1%, while insignificant changes of a similar magnitude were found for the total aerosol number. Over the time period studied, no evidence of new particle formation was found, although condensational growth in boundary layer and free troposphere in the smallest aerosol bins may have occurred. No evidence of direct or indirect aerosol effects were found in response to these very small changes in aerosol mass and number.

Whether the lack of influence from coral reef-derived DMS on the local aerosol burden was a result of unfavourable synoptic conditions or model limitations is difficult to assess. However, the close proximity of anthropogenic aerosol emissions to the inner GRB suggests that this region should not be considered a 'clean marine' environment unless under very specific conditions, therefore limiting the role that coral reef-derived DMS can play in aerosol formation and growth while these emissions continue. We suggest that the influence of anthropogenic aerosol is analysed in future work by exploring if an increase in ammonium was also found, associated with sulfate produced by a power station. While Modini et al. (2009) suggested that they had observed new particle formation in such 'clean marine' conditions, further work needs to be done to understand how often such conditions occur over the GBR before we can consider if these new particle formation events could have an influence on aerosol and weather.

This study indicates that it is more likely the small contribution of volatile sulfur compounds from the GBR contribute to aerosol growth via condensational pathways. While this is in agreement with the hypothesis presented in Jackson et al. (2020b), our results suggest that the growth in the smaller sized aerosol (Aitken mode) due to coral reef-derived DMS is still too small to have an impact on radiative or cloud processes. This finding is in agreement with the global simulation study described in Fiddes et al. (2021). We suggest future work focuses on what the influence of coral-reef derived DMS may be under 'pristine' conditions. Planned work will target the 'clean marine' period identified in the R2R campaign and consider the

downwind processes from the RVI to AIRBOX, where the airmass crosses coral reefs. As this work has shown, small increases in sulfate aerosol mass is found directly over coral reefs and how this evolves downwind is of interest. Modelling studies that test the sensitivity of influence of coral-reef derived DMS to other aerosol burdens (eg. anthropogenic, biomass burning or sea-spray) would also be of significant value. Of further interest, and perhaps yielding more significant results, would be a study conducted under pre- or post-industrial emissions conditions. Simulations such as this become particularly relevant if we consider a post-anthropogenic aerosol emissions world, in which coral reefs such as the Great Barrier Reef may already be extinct.

*Code and data availability.* RVI data including ship location, meteorology, black carbon and radon are available on the CSIRO Marlin Metadata System: https://www.cmar.csiro.au/data/trawler/. Remaining RVI and AIRBOX data will be submitted to PANGAEA in the near future, and in the mean time is available upon request. WRF-Chem namelists are available upon request and data can be made available upon reasonable request. WRF-Chem analysis was performed using the wrf-python software package (Ladwig, 2017).

## Appendix A:  WRF-Chem Chemistry, aerosol and physics set-up

**A1  Chemistry and aerosol**

The Carbon Bond Mechanism Z (CBMZ) chemical mechanism with aqueous chemistry and DMS is used in conjunction with the Model for Simulating Aerosol Interactions and Chemistry (MOSAIC) aerosol scheme. Dry deposition of gases and aerosol are turned on, as is wet scavenging (including convective wet scavenging). In-cloud chemistry, turbulent mixing and subgrid convective transport is also switched on. The FTUV (Fast Tropospheric Ultraviolet-Visible) photolysis scheme (Tie, 2003) is
used.

MOSAIC represents aerosol via a sectional approach with eight discrete size bins. For each bin, the number and mass of particles are simulated: defined by the lower and upper limit of the dry particle diameter (Zaveri et al., 2008). Particle growth is calculated in a Lagrangian manner and transfer of particles between bins is calculated using a two-moment approach (Simmel and Wurzler, 2006). Coagulation of aerosol is calculated according to Jacobson et al. (1994). Homogeneous nucleation of
$H_2SO_4$-$H_2O$ in MOSAIC is calculated via the Wexler et al. (1994) scheme. In MOSAIC, growth to Aitken mode particles (the smallest bin size in this simulation) is simulated implicitly as newly nucleated particle sizes are smaller than the smallest simulated aerosol size in the model. Heterogeneous nucleation in MOSAIC is partitioned into two schemes, treating condensation of non-volatile gases ($H_2SO_4$ and methanesulfonic acid) and condensation and evaporation of semi-volatile gases ($HNO_3$, $HCl$ and $NH_3$) separately (Zaveri et al., 2008).

MOSAIC includes 11 specific aerosol species: sulfate ($SO_4^{2-}$ and $HSO_4^{-}$), methanesulfonate, nitrate, chloride, carbonate, ammonium, sodium, calcium, black carbon, primary organic matter plus water, and treats other unspecified aerosol species as a lumped mass or through substitutions of equivalent species. Some gas phase species, including sulfuric acid and MSA, are allowed to partition to the particle phase. Atmospheric DMS chemistry is not a part of the CBMZ scheme, but was added with the development and coupling of MOSAIC (Zaveri et al., 2008). The DMS chemistry is based on that of Zaveri (1997) and
includes 11 species and 30 reactions.

The Liss and Merlivat (1986) parameterisation is used to calculate the flux$_{DMS}$ emissions in WRF-Chem. As noted in Fiddes et al. (2018) and (Fiddes et al., 2021), the Liss and Merlivat (1986) parameterisation is considered a conservative method but is believed to be the most realistic (Vlahos and Monahan, 2009; Bell et al., 2017). Sea salt emissions are calculated online via either the Gong et al. (1997) parameterisation or the Fuentes et al. (2010) adaptation which includes a large addition of
marine organic matter. All sea salt schemes have been found to overestimate sea salt emissions in WRF-Chem (eg. in Saide et al., 2012) and the Fuentes method especially so. Hence for simulations using the Fuentes high organics option, the sea salt emission was halved, as was the sea salt mass within the boundary and initial conditions.

Dust emissions are calculated via the Shao et al. (2011) scheme. Wet deposition of dust has been turned on and follows the Jung and Shao (2006) method. For daily biomass burning emissions, fire location data is provided by FIRMS (Fire Information
for Resource Management System) via the MODIS (Moderate Resolution Imaging Spectroradiometer) Collection 6 platform operated by NASA and available from https://earthdata.nasa.gov/firms. The Brazilian Biomass Burning scheme (Longo et al., 2010) calculates the respective emissions and plume rise at daily resolution. Biogenic emissions are calculated online using the

Guenther scheme (Guenther et al., 1994; Simpson et al., 1995). Anthropogenic emissions are prescribed from the Emissions Database for Global Atmospheric Research (EDGAR) V4.2, available at https://edgar.jrc.ec.europa.eu/ (European Commission Joint Research Centre and Netherlands Environmental Assessment Agency, 2012). Aircraft and volcanic emissions are not included.

## A2   Physics

All simulations use the Morrison double moment cloud microphysics scheme (Morrison et al., 2008). The RRTMG (Rapid Radiative Transfer Model for General circulation models) model for longwave and shortwave radiation is used (Iacono et al., 2008), including the Monte Carlo Independent Column Approximation method for random cloud overlap (Barker et al., 2003). Cumulus parameterisation was performed using the Grell 3D scheme, similar to the Grell-Devenyi Ensemble Scheme (Grell and Dévényi, 2002). In addition, cumulus radiation effects are switched on, allowing interaction of the radiation scheme and parameterised convective clouds (Gustafson et al., 2007). Cloud fractions are calculated using the Xu and Randall (1996) method, while cumulus and aerosol radiative feedbacks are permitted. Aerosol optical properties (Mie calculations) are approximated using the volume averaging method. The Noah Land Surface Model is used, with soil temperature and moisture in four layers, fractional snow cover and frozen soil physics (Chen and Dudhia, 2002). The boundary layer scheme used is the Mellor-Yamada-Janjic scheme (Janjić, 1994) operating in conjunction with the surface layer physics scheme, Eta (Janjic, 1996).

In WRF-Chem, with the chemistry and physics options described above, aerosol direct and indirect effects are permitted via the radiative, photolysis and cloud microphysical schemes. For direct aerosol effects, the size, number and composition of aerosol and aerosol water, refractive indices of aerosol types (based on literature) and the Mie calculations (Bohren and Huffman, 1998) update the AOD, the single scattering albedo and the asymmetry factor used in the RRTGM radiation scheme. Aerosol water has a large impact, and hence the relative humidity must also be considered when direct effects are being examined. In this work, water vapour has been nudged to the BARRA reanalysis, hence should not change significantly between experiments.

For indirect aerosol effects, the CCN number and composition is used to calculate the CDN in aerosol activation modules (Abdul-Razzak and Ghan, 2000, 2002). Activation depends on the composition and size of the particle (i.e. their hygroscopic properties), as well as the vertical and turbulent velocities of the air mass. For the first indirect effect, CDN and the cloud water mixing ratio is used to calculate the cloud particle size and effective radius, which then informs the calculation of cloud albedo. The second indirect effect is simulated within the cloud physics routines (Morrison et al., 2008), which are informed by the CDN and subsequently updates the autoconversion rate, rain mixing ratio and precipitation of the module. Lastly, for the semi-direct effect, the cloud optical properties are influenced by the absorption of solar ultra-violet and infrared fluxes altering the heating rate of cloud liquid water. It must be noted that the indirect effects can only be simulated via the microphysics and hence at non-cloud resolving scales care must be taken in the interpretation.

## Appendix B:  Field observation methods

**Table B1.** List of RVI observations used in the WRF-Chem evaluation, the institute responsible for data collection and processing, details on the instrument used and literature relevant to the data processing methods

| Field | Institute | Instrument | Methods* |
|---|---|---|---|
| $DMS_w$ | NIES/University of Tsukuba | GC-2010-FPD & EI-PTRMS | Omori et al. (2013, 2017); Kameyama et al. (2009) |
| $DMS_a$ | CSIRO | PTR-MS | NA |
| $SO_4^{2-}$, Organics, $NH_4^+$, $NO_3^-$ | QUT/CSIRO | ACSM | Fröhlich et al. (2013) |
| Black Carbon | CSIRO/QUT | MAAP | Kanaya et al. (2013) |
| N10 | QUT | CPC (TSI, Model 3772) | NA |
| CCN | CSIRO/QUT | CCNc | NA |
| Radon | ANSTO/CSIRO | Radon detector | Chambers et al. (2021) |
| Meteorology | CSIRO | Various - see CSIRO (2016) | CSIRO (2016) |
| BLH | BoM/AAD | Leosphere R-MAN510 Raman UV polarization lidar | Alexander and Protat (2018, 2019); Baars et al. (2008) |

*Refer to in text data description for full methods

**Table B2.** List of AIRBOX observations used in the WRF-Chem evaluation, the institute responsible for data collection and processing, details on the instrument used and literature relevant to the data processing methods

| Field | Institute | Instrument | Methods* |
|---|---|---|---|
| $DMS_a$ | Southern Cross University | GC-PFPD | Swan et al. (2015) |
| $SO_4^{2-}$, Organics, $NH_4^+$, $NO_3^-$ | QUT | AMS | Drewnick et al. (2005) |
| Black Carbon | CSIRO/QUT | MAAP | Kanaya et al. (2013) |
| N10 | QUT | CPC (TSI, Model 3772) | NA |
| Radon | ANSTO | Radon detector | Chambers et al. (2021) |
| Meterology | UoM | Thompson WS800 | NA |
| BLH | UoM/Chinese Academy of Sciences | Sigma Space MiniMPL-532-C Micro, | Chen et al. (2019); Xiang et al. (2019) |

*Refer to in text data description for full methods

## B1   DMS fields

$DMS_w$ observations were taken by the RVI for the duration of the voyage. Seawater samples (from between 0-5 m depth) were pumped through to the wet chemistry laboratory as part of the ship's routine underway measurements every minute. For the first half of the voyage (up until the 14th October) a Shimadzu gas chromatograph (GC-2010) (employing purge and trap gas chromatography) was used to detect $DMS_w$. After the 14th October, due to competing demands for the GC, $DMS_w$ was measured using an Equilibrator Inlet (EI) - Proton Transfer Reaction Mass Spectrometry (PTRMS) system (Kameyama et al., 2009; Omori et al., 2013, 2017). The seawater samples pumped by the ship system were flowed continuously into a 10-L glass

530

equilibrator at $1\,\mathrm{L\,min^{-1}}$. Pure nitrogen flowed from the bottom to the upper outlet of the equilibrator at $120\,\mathrm{sccm}$. Dissolved DMS was extracted into the $N_2$ gas phase and introduced into the PTRMS (Ionicon Analytik GmbH, Innsbruck, Austria). The mass signal of DMS in the sample gas was obtained at 10-s integration at 1-min intervals. The $DMS_w$ concentrations were calculated from concentrations in the sample gas extracted from the equilibrator with Henry's law constant (Kameyama et al., 2009). Comparison between the $DMS_w$ observations taken by the two techniques show excellent agreement ($r^2 = 0.999$) with high confidence ($p < 0.001$).

Accompanying the $DMS_w$ measurements on board the RVI were $DMS_a$ observations. A commercially available high sensitivity PTQRMS (Proton Transfer Quadrupole Reaction Mass Spectrometry; Ionicon Analytik GmbH, Innsbruck, Austria) was used to measure $DMS_a$. The PTQRMS sampled via a 30 m 3/8 inch ID PFA line with an inlet at the top of the mast on the foredeck 17 m above water level at a flow rate of $5\,\mathrm{L\,min^{-1}}$. The PTQRMS scanned masses from m/z 21 to 160 giving a full mass scan approximately every 10 minutes. $DMS_a$ was measured at m/z 63 corresponding to the protonated parent molecule ($C_2H_6SH^+$). The PTQRMS drift tube was operated with an applied voltage of 600V, pressure of $2.2\,\mathrm{mbar}$ (E/N = 133 Td) and a average primary ion signal (m/z 19) of 1.78 E+07 cps. Data was filtered to remove periods of instrument instability. The PTQRMS was operated with a CSIRO custom built auxiliary system which controlled whether the PTQRMS sampled VOC-free air to determine instrument background, calibration gas to determine instrument response or ambient air. Zero air measurements occurred twice daily (1 - 2 AM and 1 - 2 PM UTC) and an interpolated zero signal was subtracted from the reported ambient and calibration measurement signals. Once per day (1400 to 1500 UTC) calibration measurements occurred by diluting a certified calibration gas standard containing $997\,\mathrm{ppb}$ $DMS_a$ (Apel -Reimer Environmental Inc, Colorado, USA) (stated accuracy $\pm 5\%$) into VOC-free air. The instrument sensitivity to $DMS_a$ was 11.3 normalised $\mathrm{cps\,ppb^{-1}}$ (rel. stdev $\pm 3\%$). The minimum detectable limit (MDL) for a single 5 sec measurement at m/z 63 ($DMS_a$) was $0.029\,\mathrm{ppb}$ determined using principles of ISO 6879 (ISO 1995). Values less than the MDL were removed.

At AIRBOX, $DMS_a$ was measured using an automated gas chromatograph (GC) - pulsed flame photometric detector (PFPD). Measurements were taken every 20 minutes via an auto-sampler programmed to control the GC-PFPD. Full details on the instrumentation and data processing, including uncertainties, can be found in Swan et al. (2015). For $DMS_a$ measurements at both platforms, the closest measurement within $\pm 10\mathrm{minutes}$ to the hour was taken for comparison to the instantaneous hourly output from WRF-Chem.

## B2  Aerosol fields and radon

Mass concentrations of black carbon (BC) at AIRBOX and on the RVI were obtained with a Thermo Scientific Model 5012 multi-angle absorption photometer (MAAP) and were used to help identify periods of air contaminated by ship exhaust. The MAAP sampled through a dedicated PM10 inlet, which was heated to minimise the influence of humidity on the BC measurements (Kanaya et al., 2013). Samples were acquired at 5 s time intervals and have been averaged to a 10-minute time resolution.

An Aerodyne compact Time-of-Flight Aerosol Mass Spectrometer (AMS) provided the non-refractory chemical composition of submicron aerosol at AIRBOX (Drewnick et al., 2005). The AMS sampled through a membrane dryer (Nafion MD-700)

and a silica gel diffusion dryer which maintained sample relative humidities below 40%. Daily measurements through a high-performance particle filter were used to calculate detection limits and correct for concentrations of background air species (Allan et al., 2004). Samples were averaged to 10-minute intervals. At this time resolution, the campaign-average detection limits were $54\,\mathrm{ng\,m^{-3}}$ (Organics), $5\,\mathrm{ng\,m^{-3}}$ ($SO_4$), $4\,\mathrm{ng\,m^{-3}}$ ($NO_3$) and $46\,\mathrm{ng\,m^{-3}}$ ($NH_4$).

On board the RVI, an Aerodyne Time of Flight Aerosol Chemical Speciation Monitor (ACSM) was used to obtain chemical composition of the non-refractory submicron aerosol. A full description of its design and operation is given in (Fröhlich et al., 2013). The ACSM inlet efficiency is at a maximum for vacuum aerodynamic diameters between 100-450 nm (Jayne et al., 2000; Liu et al., 2007) and therefore the composition measurements best represent accumulation mode aerosol. The AMS sampled through a membrane dryer (Nafion MD-700) which maintained sample relative humidities below 40 %. Samples were averaged to 10-minute intervals, and at this time resolution the campaign-average detection limits were $127\,\mathrm{ng\,m^{-3}}$ (Organics), $18\,\mathrm{ng\,ng\,m^{-3}}$ ($SO_4$), $11\,\mathrm{ng\,m^{-3}}$ ($NO_3$) and $151\,\mathrm{ng\,m^{-3}}$ ($NH_4$).

At AIRBOX, the total number concentrations of aerosol with diameters larger than 10 nm were measured with a series of condensation particle counters (CPCs). Sampling was initially performed with a TSI Model 3787 CPC. This unit failed on 1st October 2016 and was replaced on the 3rd October with a TSI Model 3782 CPC. In the intervening period, aerosol concentrations were calculated from size distributions measured by a TSI 3080 scanning mobility particle sizer (SMPS). The SMPS was operated with a TSI Model 3782 CPC and a custom-made differential mobility analyser, which allowed measurement of aerosol with diameters from 11-600 nm. From 11th October 2016 onwards, a newly calibrated TSI Model 3772 CPC was operated in parallel with these instruments, providing reference measurements that were used to correct the counting efficiencies of SMPS and Model 3782 CPC. In turn, the counting efficiency of the Model 3787 CPC was calibrated against the Model 3782 based on co-located measurements taken at the beginning of the campaign. The measurements were averaged to a 3-minute time resolution.

On board the RVI, total number concentrations of aerosol with diameters larger than 10 nm were measured with a TSI model 3772 (TSI, Shoreview, MN) CPC. Aerosol size distributions were measured at diameters from 14 to 685 nm the using a TSI 3080 SMPS. The SMPS was operated with a TSI Model 3772 CPC and a TSI 3081 differential mobility analyser, with a sheath flow of 3 L/min and aerosol flow of 0.3 L/min.

The number concentration of CCN was measured on the RVI using a continuous-flow streamwise thermal-gradient CCN counter (CCNc, model CCN-100, Droplet Measurement Technologies, Longmont,CO, USA). The instrument was configured to run continuously at 0.5 % supersaturation and the flow rate was set to 0.5 L/min. The CCNc and CPC measured from approximately the same point on the sampling line. At AIRBOX, CCN measurements were unavailable due to instrument failure.

In addition, atmospheric radon-222 concentrations were measured both at AIRBOX and on the RVI using dual-flow-loop two-filter atmospheric radon detectors. Radon concentrations have been shown to be an accurate, independent measure of residual terrestrial influence within an airmass. Radon can subsequently be used to determine if an airmass has a marine or terrestrial origin, and can be satisfactorily used to detect 'baseline' airmass at locations such as Cape Grim, Tasmania, Australia (Chambers et al., 2021). While at Cape Grim, the baseline radon concentration is considered to be $80\,\mathrm{mBq\,m^{-3}}$ or

below, on board the RVI and at AIRBOX, terrestrial influence over the airmasses was much higher given their coastal location, prevailing wind directions and the fact that coral atolls, when exposed, are also a source of radon. For the RVI, a threshold of 300 mBq m$^{-3}$ was used to determine marine from terrestrial airmass. At AIRBOX no satisfactory threshold of radon could be determined to separate marine and terrestrial influences.

## B3  Meteorology

On board the RVI, meteorological observations were taken as part of the routine observations. Observations are available at 5 or 10 second, or 5 minute intervals. The five minute interval has been used in this study. Where port and starboard observations were available, an average over the two was taken. These observations have been processed by the Marine National Facility and can be downloaded from the Marlin Metadata System (CSIRO, 2016), where more information can also be found. At AIRBOX, wind speed and direction, relative humidity, temperature and pressure observations were taken using a Thompson WS800 weather station. Observations were taken every 15 seconds. These observations have undergone basic quality control.

A Leosphere R-MAN510 Raman UV polarization (RMAN) lidar operating at 355nm collected profiles of attenuated backscatter and depolarisation throughout the voyage, which were then processed following the techniques described in Alexander and Protat (2018) and Noh et al. (2019). Advancements to the lidar processing algorithm for this campaign include the use of optically-thick liquid non-precipitating stratocumulus clouds for calibration of the lidar (O'Connor et al., 2004) and the initial extraction of the brightest cloud features in the co-polarised channel and subsequent assignment of cloud phase (Alexander et al., 2021; Hu et al., 2009) before the fainter cloud pixels are processed. Additionally, hydrometeor and aerosol pixels are flagged, using empirically-determined campaign-specific thresholds of molecular backscatter and variance, followed by the removal of spurious signals using a region-of-interest analysis. This new step results in a much larger detection of ice virga as the crosspolarisation channel data are also utilised. Liquid precipitation, include that which reaches the surface, are readily detected in this additional step. This improved cloud-precipitation-aerosol detection algorithm allows us to extract clear-air profiles (where hydrometeors are absent) which are then used to detect the boundary layer height (BLH) using the wavelet covariance transform method as described by (Baars et al., 2008). Alexander and Protat (2019) showed that this method agreed closely with co-located radiosonde measurements of the boundary layer altitude over the Southern Ocean.

The BLH at AIRBOX was estimated from observations by a scanning Mini Micro Pulse lidar -532-C Micro (MiniMPL) System (Sigma Space Corporation, Lanham, MD, USA). The MiniMPL operates at 532 nm wavelength and retrieves elastic aerosol backscatter every 10-20 seconds between 30-9,990 m. More information about the MiniMPL, including the set-up, calibration and data processing can be found in Chen et al. (2019). To detect the BLH an extension of the gradient method of lidar backscatter was employed, which is detailed in Xiang et al. (2019).

## Appendix C:  Additional evaluation plots

In this section we provide additional plots further evaluating the model against observations. These plots have been provided as a reference and will not be discussed.

635   *Author contributions.*  SF completed the WRF-Chem simulations, analysis and the initial draft of this manuscript. SU developed the initial model setup and provided advice as to the specific setup requirements of this study. MTW and SU helped guide the analysis and contributed significantly to the revisions of this manuscript. RS and TL provided advice and guidance on the direction of this study and contributed to the revisions of this manuscript. Remaining authors contributed observational data to this study and to the revisions of this manuscript.

*Competing interests.*  The authors declare that they have no conflict of interest

640   *Acknowledgements.*  This study and it's authors were supported by the ARC Discovery Project: Great Barrier Reef as a significant source of climatically relevant aerosol particles (DP150101649). SLF would like to thank P. J. Rayner and his research group for their helpful discussions and D. McConnell for providing the locations of power generators across Queensland. SLF was supported by the Australian Research Council (ARC) Centre of Excellence for Climate System Science (CE110001028) and the Australian Antarctic Program Partnership. TPL is supported by the Australian Research Council (ARC) Centre of Excellence for Climate Extremes (CE170100023). Financial support was

645   given to HT and YO by Grant-in-Aid for Scientific Research (15H01732 and 17KK0016) from the Ministry of Education, Culture, Sports, Science and Technology, Japan. This research was undertaken with the assistance of resources and services from the National Computational Infrastructure (Project q90 and w40), which is supported by the Australian Government. SLF was supported by the Australian Government Research Training Program Scholarship.

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

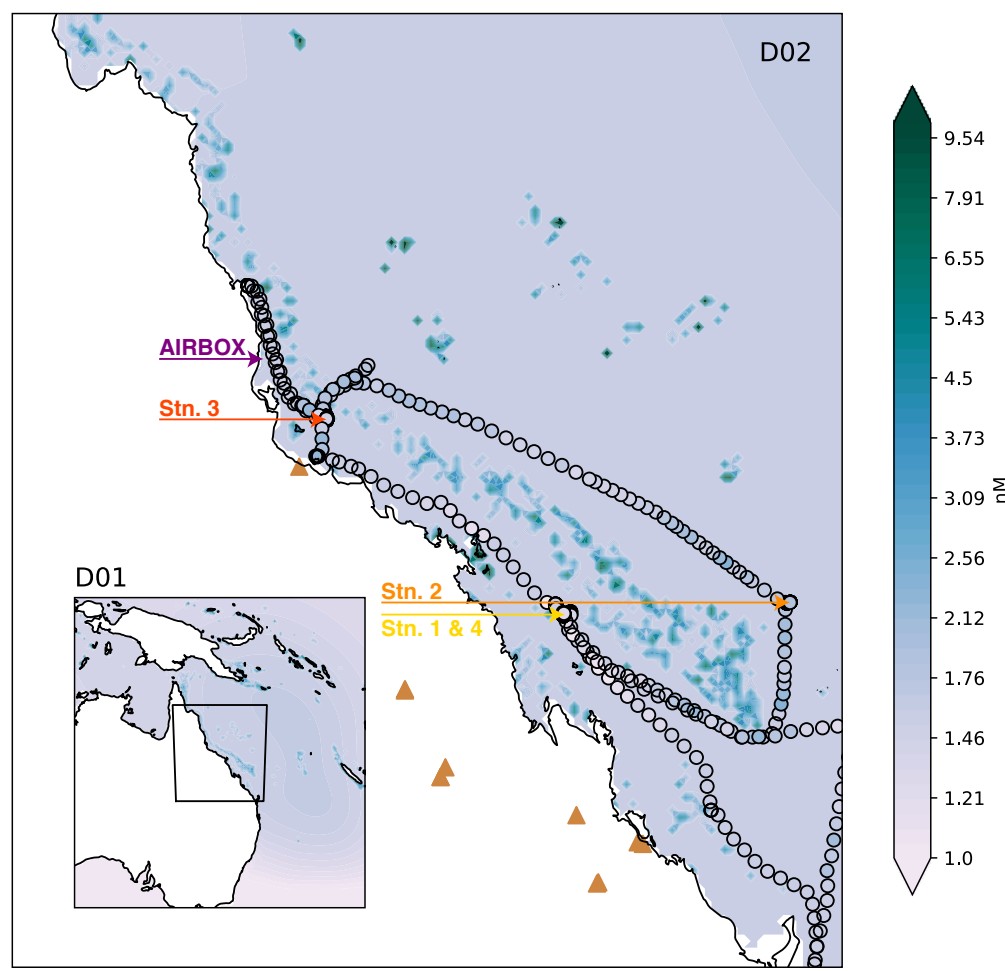

**Figure 1.** Domain two (D02) of the WRF-Chem simulations, with the outer domain (D01) inset. The colours represent the DMS surface water concentrations for the simulations including coral reef-derived DMS (note the log scale). The RVI ship track is shown by black outlined circles, also coloured by DMS observations. The locations of the RVI stations and AIRBOX are shown by coloured text and arrows. The brown triangles indicate where fossil fuel or biomass burning power generators are located

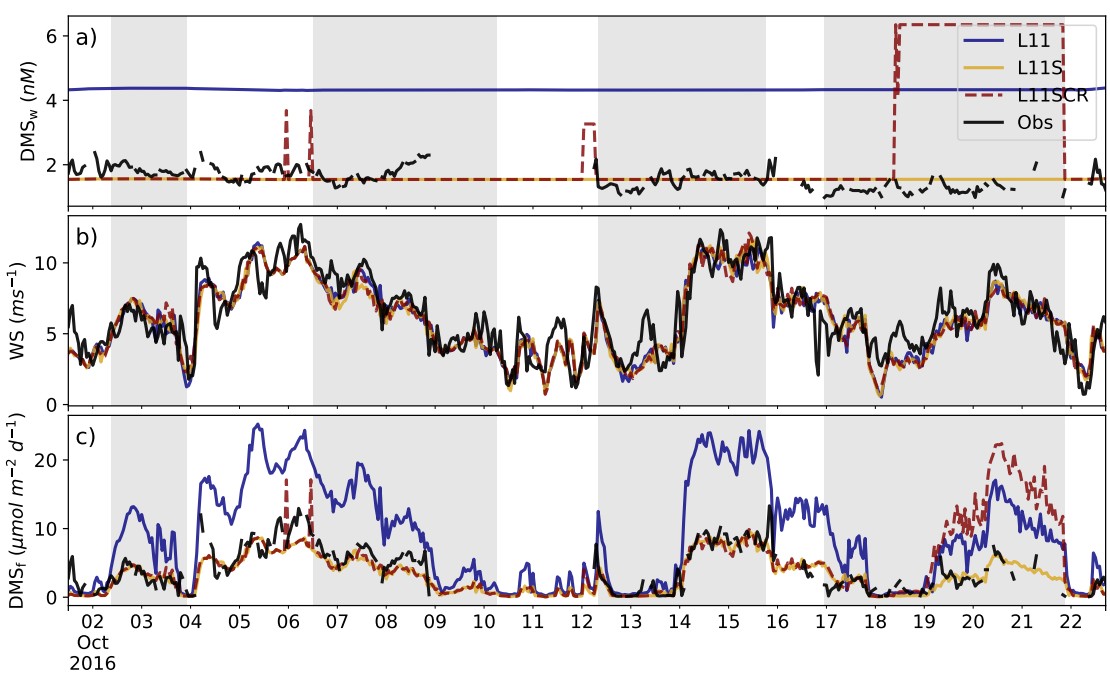

**Figure 2.** RVI observations (black) and model estimates (coloured) of a) DMS$_w$ (nM), b) wind speed (m s$^{-1}$) and c) the flux$_{DMS}$ ($\mu$ mol m$^{-2}$ day$^{-1}$). Note the flux$_{DMS}$ is calculated from observations using the Liss and Merlivat (1986) parameterisation. The three WRF-Chem simulations are L11 (blue), L11S (yellow) and L11SCR (red, dashed). They grey shading from left to right indicates when the ship was at station locations 2, 3.1, 3.2 and 4.

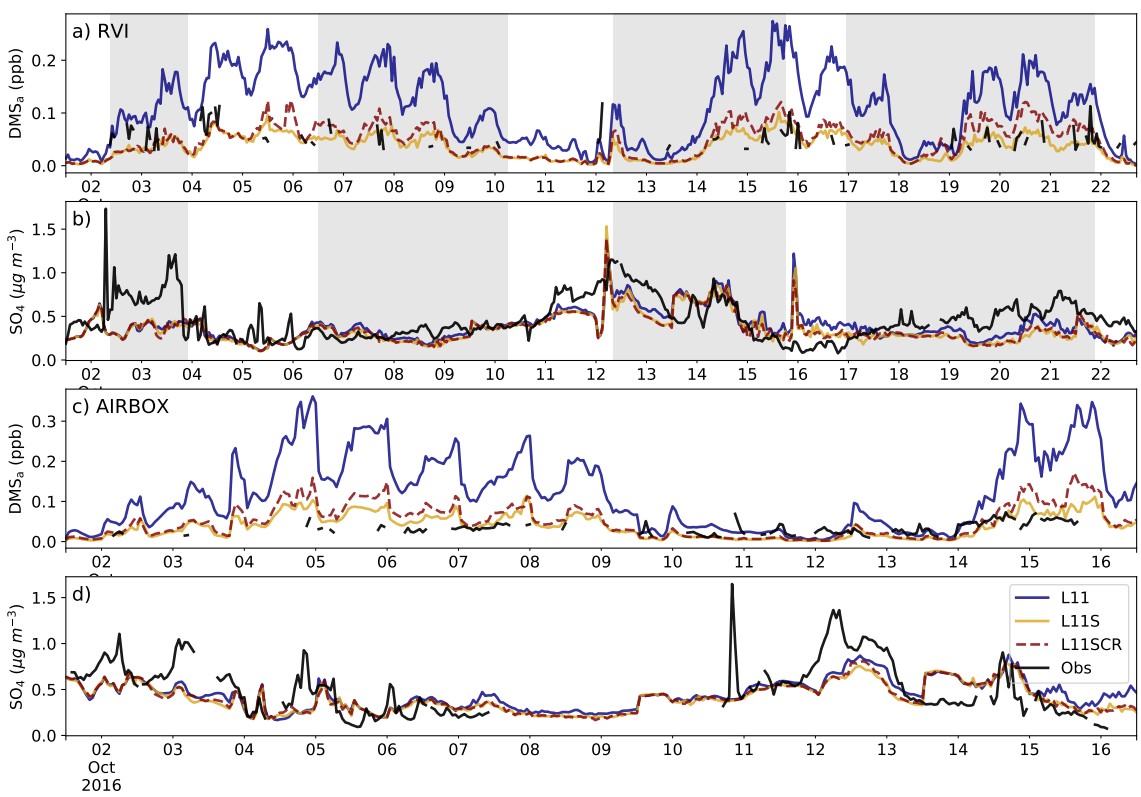

**Figure 3.** Observations (black) from the RVI (a,b) and AIRBOX (c,d) and corresponding model estimates (coloured) of a & c) $DMS_a$ (ppb), b & d) $SO_4^{2-}$ aerosol mass ($\mu g \, cm^{-3}$). The three WRF-Chem simulations are L11 (blue), L11S (yellow) and L11SCR (red, dashed). They grey shading from left to right indicates when the ship was at station locations 2, 3.1, 3.2 and 4. Note that the dates in a, b and c, d are not the same.

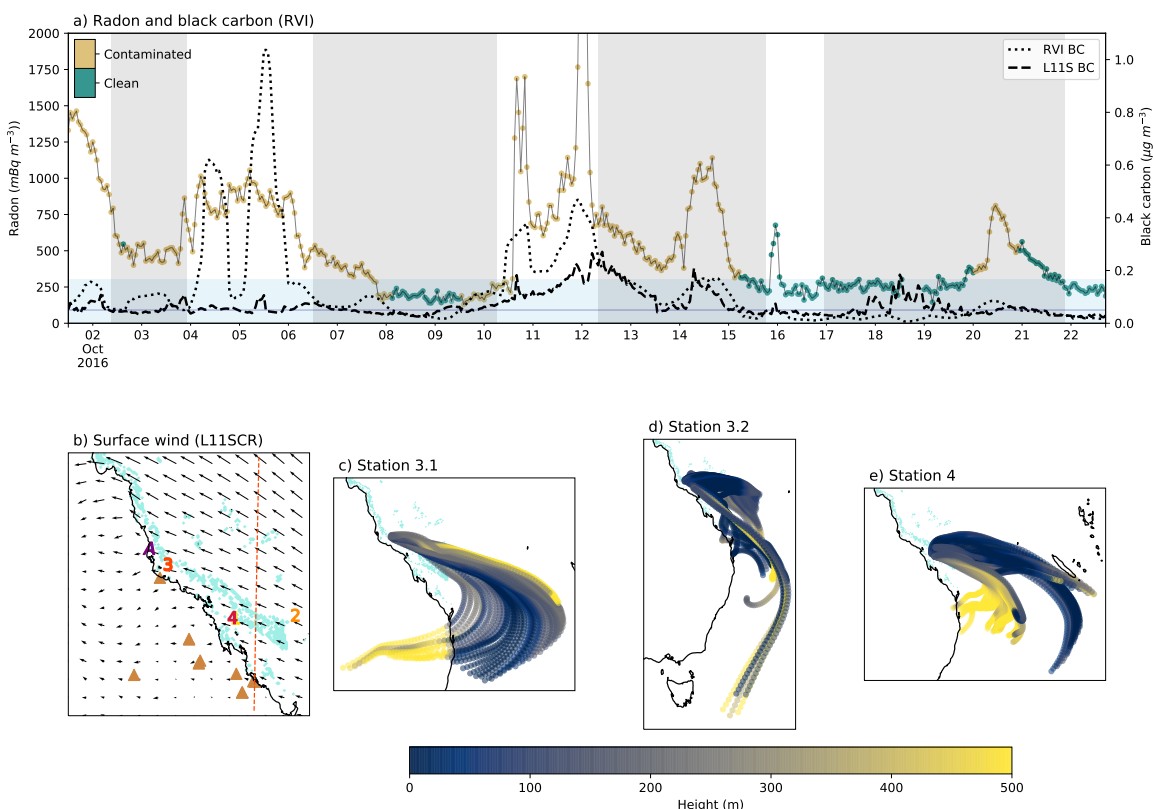

**Figure 4.** a) Radon concentrations from the RVI and coloured by the ship exhaust flag (left axis). Black carbon concentrations (right axis) from the RVI (dotted line) and L11S (dashed line). The light blue shading indicates when radon was below $300\,\mathrm{mBq\,m^{-3}}$ and the dark blue line indicates black carbon below $0.05\,\mu\mathrm{g\,m^{-3}}$. The grey shading represents ship stations 2, 3.1, 3.2 and 4. b) the surface winds from L11SCR. Coral reefs are shown in light blue. Numbers indicate RVI observation stations 1-4 coloured yellow through to red and the purple A indicates the AIRBOX location. The brown triangles indicate where fossil fuel power generators are located and the dashed orange vertical line indicates where a transect was taken for vertical profile analysis. c-e) HYSPLIT back trajectories for every two hours while the ship was at station 3.1, 3.2 and 4 respectively, coloured by height.

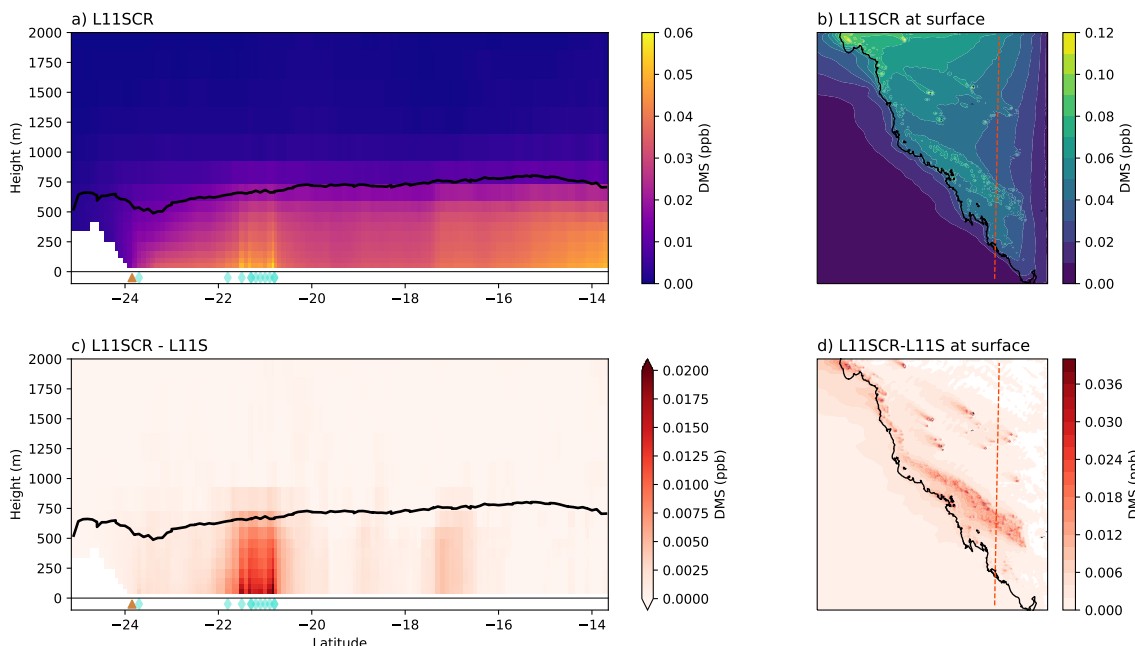

**Figure 5.** Vertical transects (a,c) and surface means (b,d) of DMS$_a$ for L11SCR (a,b) and the difference between L11SCR and L11S (c,d). In a and c the black line indicates the mean simulated boundary layer height, while the brown triangle shows the location of the Gladstone power station and the blue diamonds show grid points within which coral reefs are found.

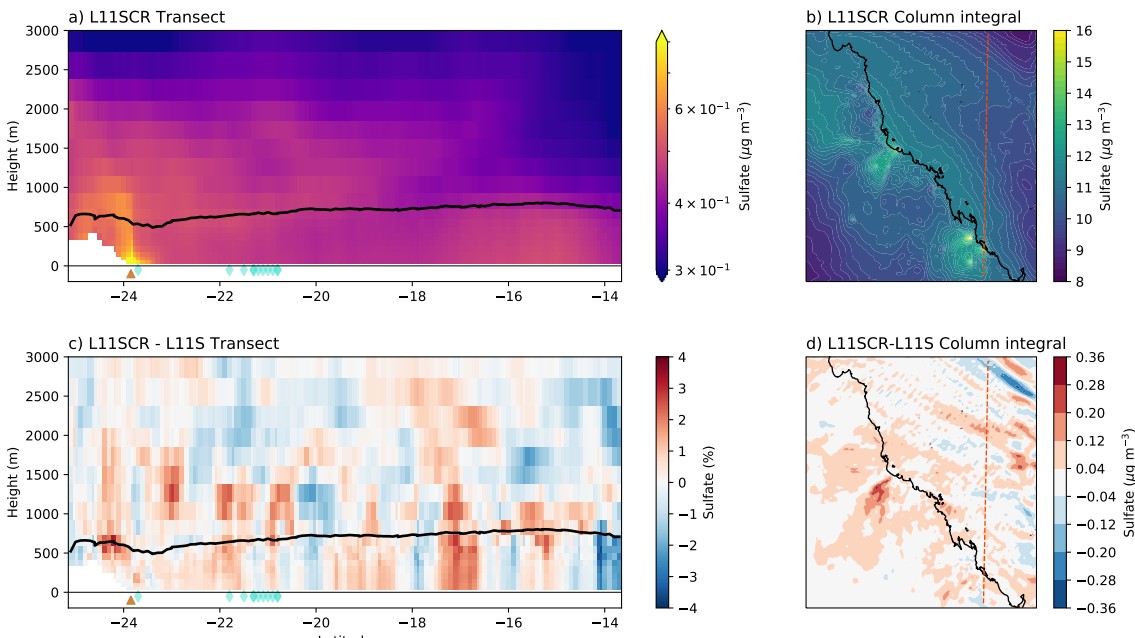

**Figure 6.** As for Figure 5 but for total sulfate aerosol mass.

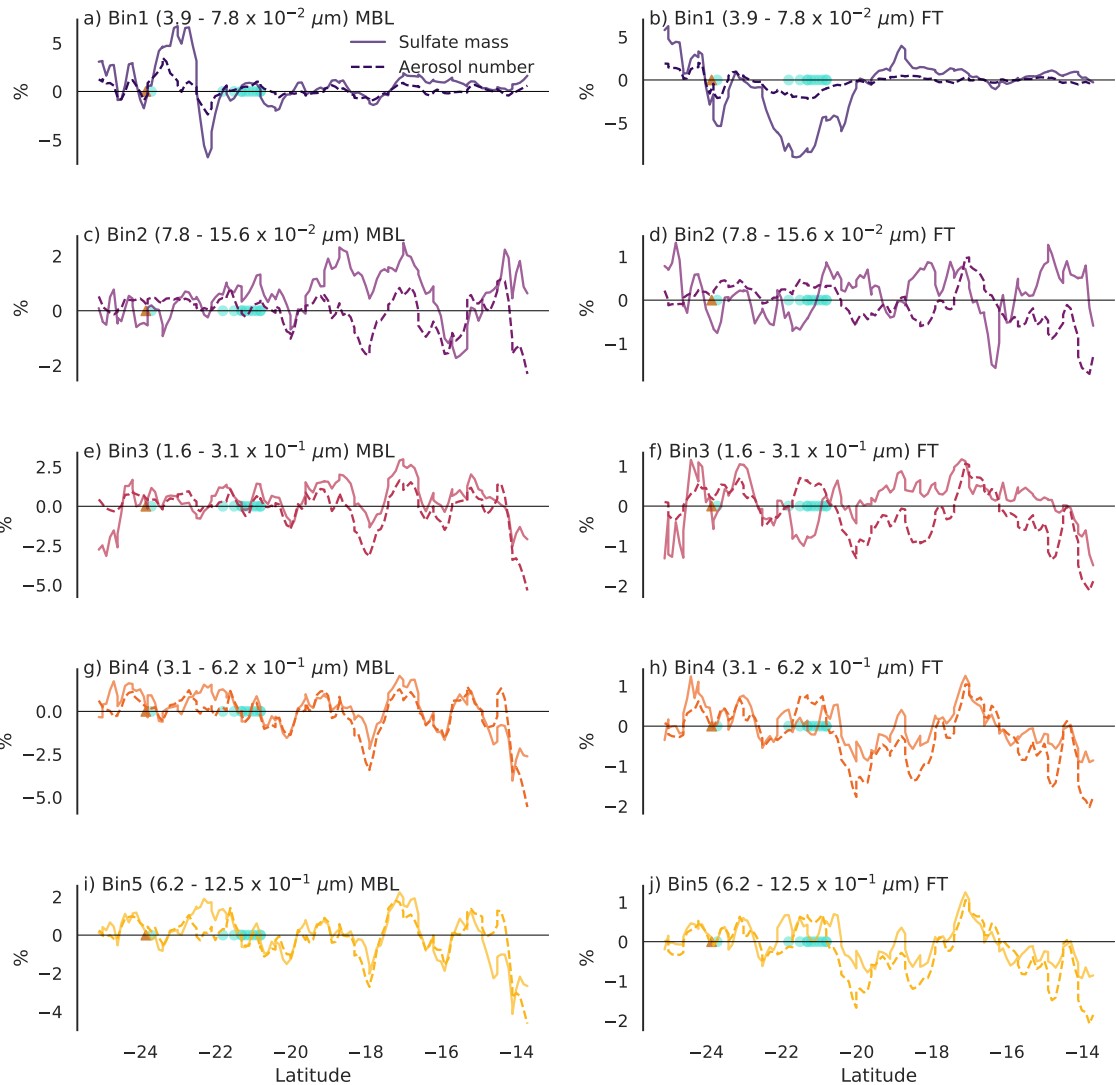

**Figure 7.** Percentage changes (L11SCR-L11S) along the vertical transect of total boundary layer column (left) and free troposphere column (right) sulfate mass (solid line) and total aerosol number (dashed line) for the five smallest bin sizes representing the Aitken mode (bin 1) and accumulation mode (bins 2-5). The brown triangle shows the location of the Gladstone power station and the blue circles show grid points within which coral reefs are found.

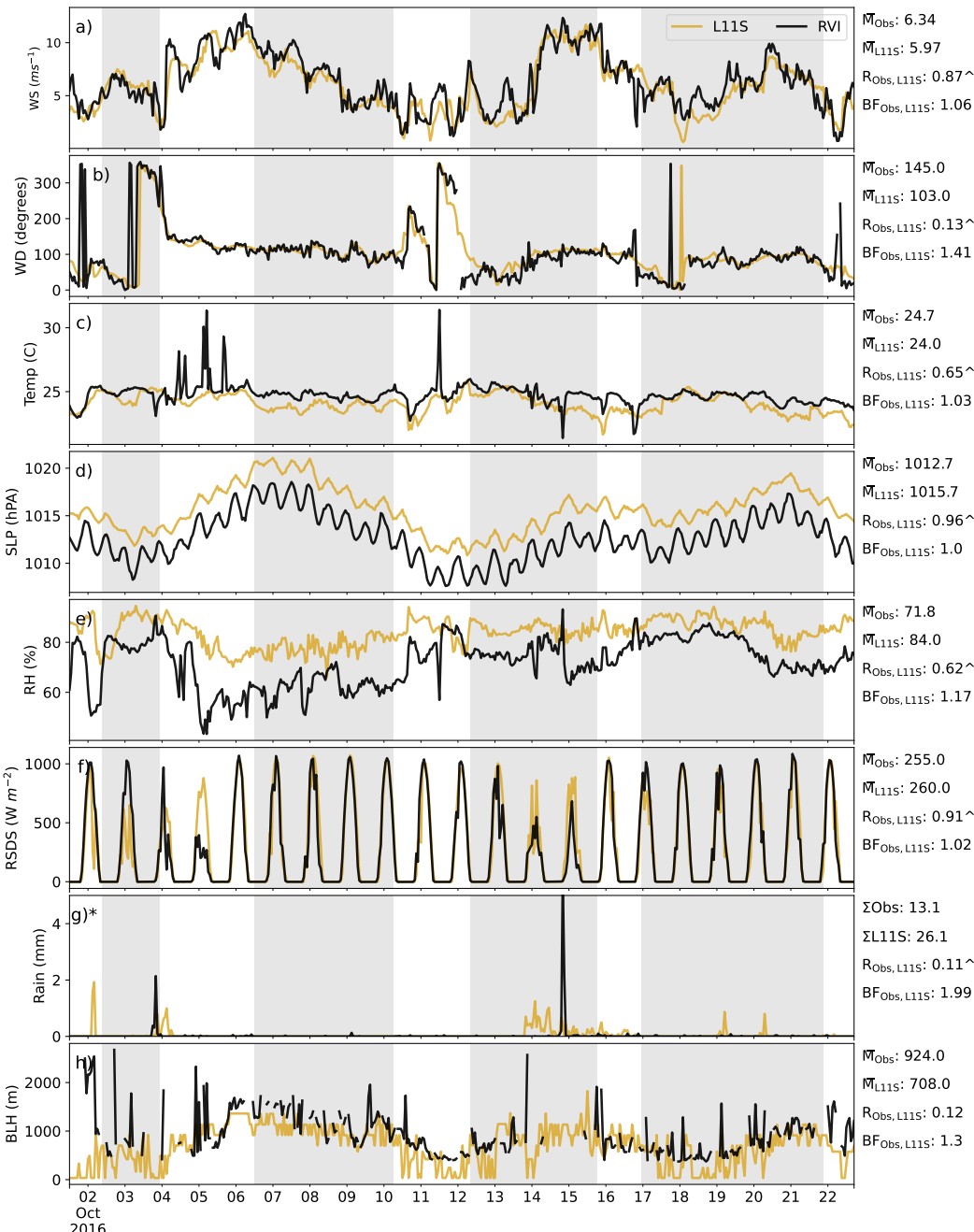

**Figure C1.** Timeseries along the RVI track (in UTC) in descending order: wind speed, wind direction, temperature, sea level pressure, relative humidity, surface incoming radiation, rainfall and boundary layer height, comparing the L11S WRF simulation (yellow) to observations (black). Summary statistics are shown for the observations and L11S including the mean ($\bar{M}$) (or sum $\Sigma$), the correlation (R) and the normalised mean bias factor. ^ indicates where the R values are significant to the 95th percentile. The shaded grey areas in a-h) represent periods when the ship was at station. The y-axis in g) has been limited to 5 mm for clarity. Observations in h) have been filtered for when cloud was detected and subsequently excluded for the relevant statistics

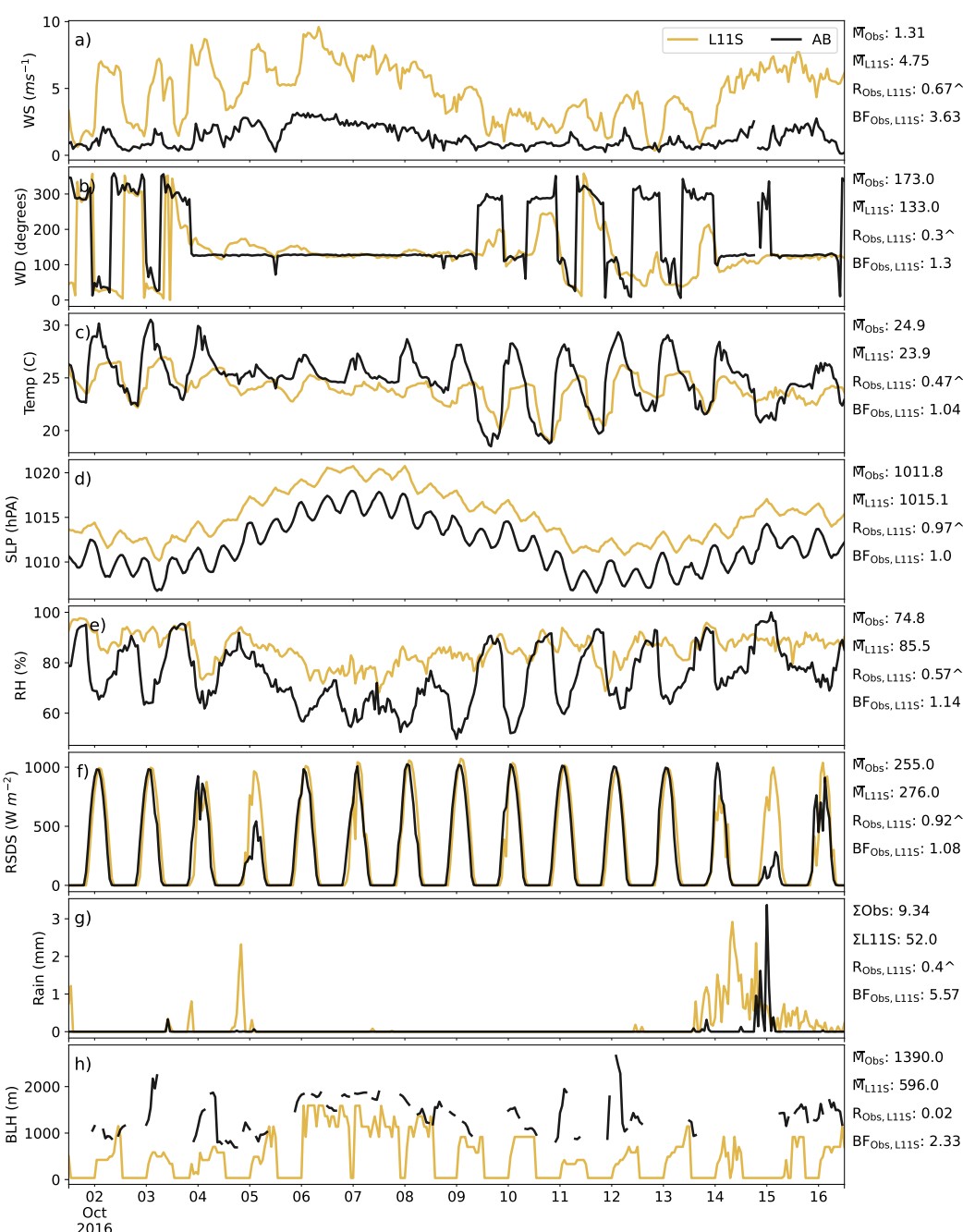

**Figure C2.** Timeseries for AIRBOX (in UTC) in order of top to bottom of wind speed, wind direction, temperature, sea level pressure, relative humidity, surface incoming radiation, rainfall and boundary layer height, comparing the L11S WRF simulation (yellow) to observations (black). Summary statistics are shown for the observations and L11S including the mean ($\bar{M}$) (or sum $\Sigma$), the correlation (R) and the normalised mean bias factor. ˆ indicates where the R values are significant to the 95th percentile.

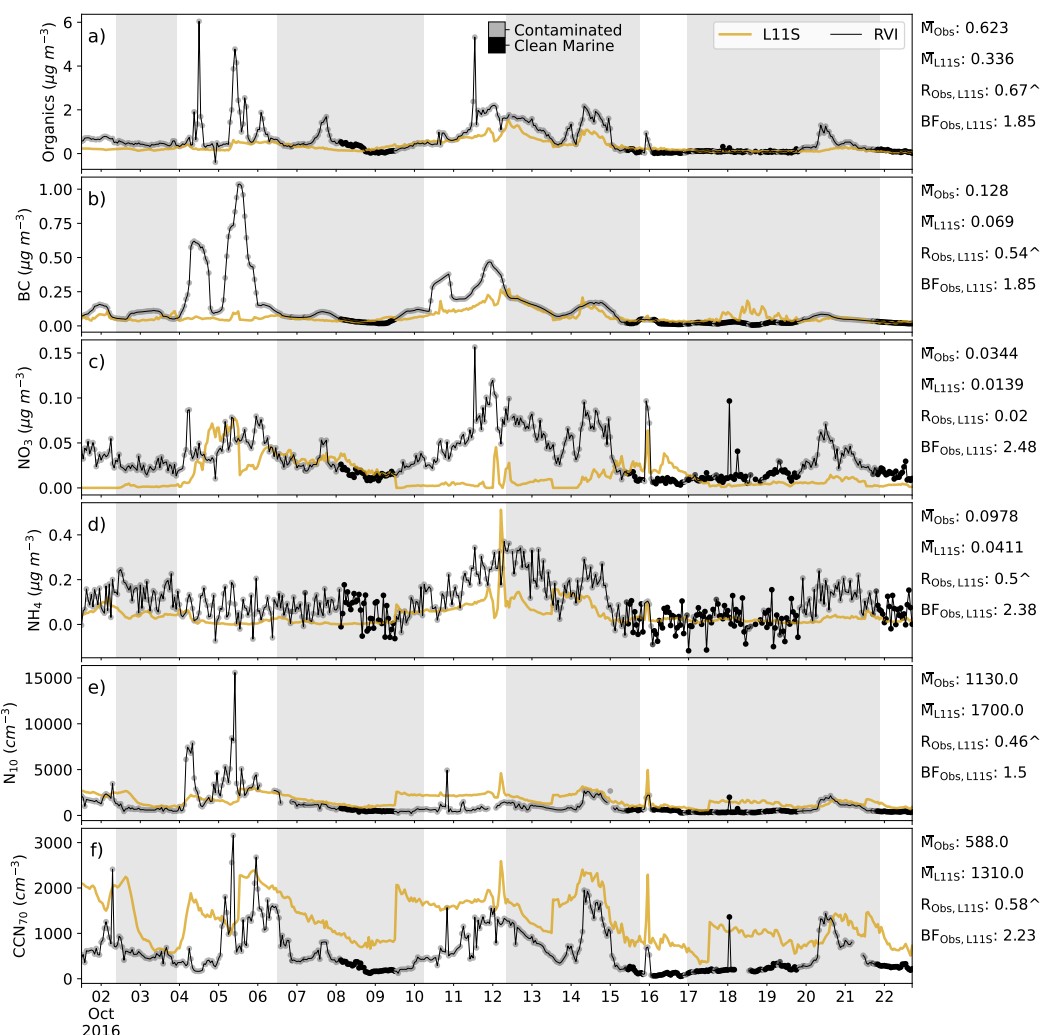

**Figure C3.** Timeseries along the RVI track in order of top to bottom of surface organic, BC, $NO_3$ and $NH_4$ aerosol mass, aerosol number greater than 10 nm ($N_{10}$) and cloud condensation nuclei greater than 70 $\mu$m ($CCN_{70}$), comparing the L11S simulation (yellow) to the observations (black). Summary statistics are shown for the observations and L11S including the mean (M), the Pearson correlation (R) and the normalised mean bias factor. î̂ndicates where the R values is significant to the 95th percentile. The shaded grey areas in represent periods when the ship was at station. Observations in the timeseries are flagged by grey dots when then airmass was considered to be influenced by exhaust, terrestrial airmass or both (contaminated), including a log flag. Flagged values have been excluded from the statistics.

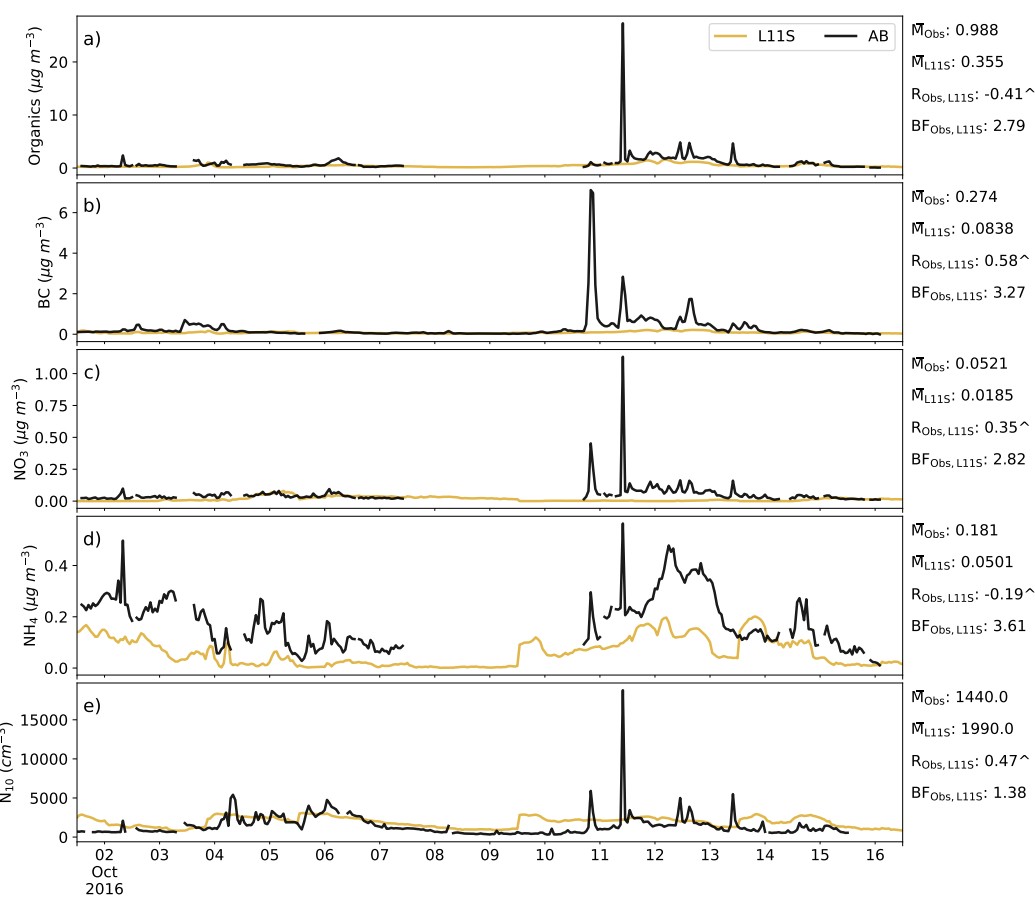

**Figure C4.** Timeseries for AIRBOX in order of top to bottom of surface organic, BC, $NO_3$ and $NH_4$ aerosol mass and aerosol number greater than 10 nm ($N_{10}$), comparing the L11S simulation (yellow) to the observations (black). Summary statistics are shown for the observations and L11S including the mean (M), the Pearson correlation (R) and the normalised mean bias factor. îndicates where the R values is significant to the 95th percentile