# Peer review of "The contribution of coral reef-derived dimethyl sulfide to aerosol burden over the Great Barrier Reef: a modelling study"

_Atmospheric Chemistry and Physics, 2021_

## Author Comment (AC1)

**Response to Reviewer 1**

Many thanks to this Reviewer for their encouraging comments. We have now included a detailed discussion section in this work to help address some of the concerns raised by this Reviewer. While further simulations are not within our current capabilities, we believe that our changes and additions to the text address the main concerns within this work, in particular surrounding the model set-up. We have further made sure to discuss the results of this work in comparison to our previous work in more detail. Our detailed response can be found below.

**Major comments**

Is this actually the first paper documenting the field campaign, or using measurements from it? If so, the authors could make more of the novelty of these results and put the DMS measurements and field campaign into a broader context.

This is the first published paper to analyse the results from both the RVI and AIRBOX campaign (there has been one looking at results from AIRBOX - Chen et al. 2018). We have not included detailed context of the campaign in this paper as this is currently being written up by H. Trounce et al. (2022), which will present a broader overview of the data. We have however, now included greater emphasis on the novelty of this data in the introduction and methods sections.

Line 88: 'We evaluate WRF-Chem against new and novel observations from a major field campaign undertaken in the austral spring of 2016: 'GBR as a significant source of climatically relevant aerosol particles', nicknamed 'Reef to Rainforest' (R2R)'

Line142: 'The 2016 major field campaign, R2R, aimed to determine if marine aerosol produced by the GBR could affect CCN concentrations, cloud formation and subsequently the hydrological cycle, providing essential observational evidence for assessing DMS-climate interaction. A leading motivation for this field campaign came from observations by Modini et al. (2009). A selection of the data from this campaign is used in this work to evaluate the WRF-Chem model.'

Line 152: 'While a subset of AIRBOX data has been described previously in Chen et al. (2018) (including lidar, aerosol, trace gas and meteorology data), this is the first work to use the new and novel dataset from across the R2R campaign. However, we note that an overview paper on this campaign is currently in preparation (Trounce et al. 2022).'

A fuller description of the meteorology during the case study would be helpful. L250/Figure 5: is the boundary layer height simulated or observed? Were there radiosondes you could use for evaluation of the vertical temperature profile? Do the simulations get the boundary layer height right? Is there a strong inversion that distinguishes the boundary layer from the free troposphere? If so can you explain the lack of clear variation of sulfate aerosol mass across the boundary layer on the left of Figure 6? If not, why split up the boundary layer and free troposphere in Figure 8?

The boundary layer height in Figure 5 (and others) is the simulated BLH. We do have BLH determined from lidar observations, which show the model performed better than expected over the ocean, but poorly at AIRBOX. We have now included in the Appendix further plots evaluating the meteorology of the both sites as a reference (including a description of the data from Line 615). We have not included a detailed discussion of these meteorological evaluations, as we believe the plots and statistics speak for themselves, and we were satisfied with the model performance in this respect, due to the nudging. We did not have radiosondes for this campaign.

Line 279: 'We note that the model captures the boundary layer height along the RVI track relatively well (Appendix Figure C1), with some exceptions due in part to the model and in part to miss-identification of cloud and a 500m depth limitation by the lidar. At AIRBOX, the boundary layer height is not as well captured (Appendix Figure C2), in agreement with Chen et al. (2019). This result also agrees with previous work that indicates the Mellor-Yamada-Janjic (MYJ) boundary layer scheme underestimates the boundary layer height the most in coastal marine areas, which then improves further offshore Rahn & Garreayd (2010)' We have split the results by the BLH because Fiddes et al (2021) found important changes in nucleation model aerosol in the free troposphere. We recognise that the delineation at BLH is not seen in the sulfate aerosol. Nevertheless, we chose to split the analysis in order to detect if there was any indication of new particle formation in the free-troposphere occurring, as opposed to boundary layer particle formation.

Line 172: 'Further characterisation of airmasses has been performed by splitting WRF-Chem aerosol into boundary layer and free troposphere masses. The simulated boundary layer height was used. This was done to explore if any specific changes to aerosol could be found in either airmass, in particular due to the differing nucleation processes that occur at the two different levels, and following from Fiddes et al. (2018) who found some impact on nucleation rates in the free troposphere in response to perturbations of coral-reef-derived DMS.'

Line 289: 'We note that the same delineation between the boundary layer and free troposphere found in the  $DMS_a$  plots is not seen in the sulfate aerosol mass.'

Line 298: 'We have separated the two atmospheric profiles as Fiddes et al. (2021) noted larger changes in the free troposphere in response to perturbed coral reef DMS than in the boundary layer.'

Uncertainties and shortcomings of the study are only briefly discussed in the conclusions of the paper. I recommend a separate section with some more quantitative analysis and sensitivity studies.

We have now added a new 'Discussion' on Line 326 section to expand our discussion on the uncertainties and shortcomings of this work, especially with respect to the remaining comments from this Reviewer, which are addressed below. We note that it is not possible to provide more simulations for sensitivity studies for this work.

In particular, nucleation mechanisms and aerosol microphysics are relevant to the author's comments in the introduction about boundary layer nucleation events from coral, and should be discussed in more detail. Could model shortcomings explain why the authors' simulations did not show up any boundary layer nucleation events?

It is possible that model short-comings may be the cause of the lack of BLN events. We have addressed this in the new discussions section (from line 377). See the responses to the next two comments for details.

The authors correctly point out a major shortcoming of the model they use, its crude representation of the aerosol size distribution, at line 341. This could be discussed in more detail. The uncertainties associated with what happens below 40nm (or even much smaller aerosol sizes) in an aerosol model are discussed by Lee et al (2013; https://gmd.copernicus.org/articles/6/1221/2013/gmd-6-1221-2013.html) and Blichner et al (2021 https://acp.copernicus.org/preprints/acp-2021-151/). The configuration of WRF- chem with more size bins, for example as used by Matsui et al (2011, https://doi.org/10.1029/2011JD016025) or Zhao et al (2020 https://www.pnas.org/content/117/41/25344) or the regional configuration of the Unified Model with GLOMAP aerosol as per Gordon et al (2018 https://acp.copernicus.org/articles/18/15261/2018/) would have resolved aerosols down to around 3nm in size as in Fiddes et al (2021). I think the authors need to explain in the paper why these models were not used (is the chemistry in UM-UKCA too simple for example? Are the configurations of WRF-chem with more bins too expensive?), and/or quantitatively compare the size distributions from WRF-chem with the size distributions from GLOMAP in Figure 7 of Fiddes et al (2021) in the relevant area.

We have now added a new 'Discussions' (Line 326) section to address these points and those below explicitly. We thank the Reviewer for their literature suggestions, they have proved most useful for our discussions. We have addressed each point above in the new Section and will not repeat them here for efficiency. We will state however, that our decision not to use a more fully resolved aerosol size distribution was in part due to the selection of a model configuration based on methods of a paper we believed to have had similar goals to ours. Unfortunately, we no longer have the capability to re-run these simulations. Nevertheless, we believe that our results are robust and have included our reasoning in the Discussion. We believe that this does not impact the overall standing of our results. In response to the Reviewer's final point, we have not done a quantitative comparison of size distributions with those from our previous work, in part because of the significant differences in resolution and time (previous work was run over the 2000s, at 1.25x1.875 degrees horizontal resolution). We have, however, made note in our discussion of the broader differences in the nucleation bin/mode sizes.

The section on nucleation pathways did not refer to the substantial corpus of prior work on the mechanisms for, or observations of, atmospheric new particle formation. Mechanisms were also not discussed in detail in the introduction. The brief discussion on nucleation presented at line 346 in the conclusion could be much more comprehensive and presented earlier. I think the nucleation scheme of Wexler et al (1994) the authors use is the parameterization of Jaeckel-Voirol and Mirabel (1989), which is based on classical nucleation theory. Zaveri et al (2008) points out that no ternary nucleation scheme is included in WRF-chem (the authors' comment on that in this paper could be easily misconstrued). Therefore, there is no participation in nucleation of stabilizing compounds which could substantially increase new particle formation rates, such as ammonia, amines or methanesulfonic acid (e.g. Brean et al 2021 https://www.nature.com/articles/s41561-021-00751-y), all of which are likely present at some level near the Australian coast. These stabilizing mechanisms are especially important at high temperatures where molecular clusters readily evaporate. Therefore, the importance of boundary layer DMS to marine CCN number concentration may be underestimated. Or, indeed, the concentration of CCN could be overestimated if there are too many nuclei acting as sinks for vapors, and therefore the particles cannot grow large enough to act as CCN- see Sullivan et al (2018; https://www.nature.com/articles/s41612-018-0019-7). I note that Sullivan et al also uses WRF-chem with additional size sections. Maybe the authors would find the reviews of new particle formation mechanisms by Lee et al (2019 https://agupubs.onlinelibrary.wiley.com/doi/10.1029/2018JD029356) and models by Semeniuk and Dastoor (2018; https://doi.org/10.1016/j.atmosenv.2018.01.039) and papers cited therein helpful to expand their discussion. I think some sensitivity studies where the nucleation mechanism and/or treatment of nucleation-mode microphysics is varied, and some discussion of the associated uncertainties, may be interesting additions to the paper.

We again thank the reviewer for their literature suggestions for this section. We have now included in our new 'Discussion' section a brief discussion on the complexity of new particle formation (from Line 361), including the fact that many observed pathways are not simulated by the parameterisation used in this work, and that coastal, tropical marine nucleation processes are far from being understood.

We have not included a major discussion on nucleation mechanisms in the introductions of this work, as for one, a lot of this is covered in the new Discussion section and two, we do not want the main point of this study to be on nucleation processes that we do not resolve, as important as they are.

Furthermore, unfortunately, we are not in a position to run sensitivity tests for this work on nucleation parameterisations. We also note that while the nucleation mechanisms in this work are essential to understand, this studies primary focus is on the role of DMS in the climate system. We have tried to emphasis this point more carefully throughout the text.

We also note that due to the addition of the new Discussion section, some parts of our conclusion have been either removed or moved, as they are now dealt with in the new section.

**Minor comments**

L23 missing citation, perhaps to Merikanto et al (2010)?.

Yes, that's the one! Have fixed.

L196 correlations between what and what? Presumably observations and model, but please specify...

Have specified correlations are between observations and model.

L246 how DMS has changed because of what? Presumably because of including the DMS source from coral, but please specify. Also should reiterate that these changes are (if I am not mistaken) simulated

changes, not observed changes..

We have clarified this: DMS has changed in response to the simulated coral reef perturbations. And yes, these changes are all simulated, we have made this clear.

L275 a decrease in sulfate compared to what? The area closer to the coast? Please specify..

We have clarified that this is compared to the simulation without coral-reef-derived DMS.

Appendix: Should explain mechanisms for aerosol-cloud interactions more carefully. How is the second indirect effect parameterized?.

We have now included greater explanation of aerosol-cloud interactions in the Appendix.

Line 528: 'For the first indirect effect, CDN and the cloud water mixing ratio is used to calculate the cloud particle size and effective radius, which then informs the calculation of cloud albedo. The second indirect effect is simulated within the cloud physics routines (Morrison et al. 2008), which are informed by the CDN and subsequently updates the autoconversion rate, rain mixing ratio and precipitation of the module. Lastly, for the semi-direct effect, the cloud optical properties are influenced by the absorption of solar ultra-violet and infrared fluxes altering the heating rate of cloud liquid water. It must be noted that the indirect effects can only be simulated via the microphysics and hence at non-cloud resolving scales care must be taken in the interpretation.'

Figure 7 the blue diamonds look like circles to me.

Have corrected the text.

**Response to Reviewer 2**

We thank Reviewer 2 for their positive comments and constructive feedback on our manuscript. We have now included additional evaluation plots for reference in the Appendix to help validate the model, however, we have not performed any additional simulations to test the sensitivity of the model to various other aerosol sources. We have addressed each comment below.

**Major comments**

1, More model validations are needed in this study. The authors used novel measured DMSa, SO4, and BC to validate the WRF-CHEM simulations, but all the data points are over the ocean left the Australian continent unaccounted. As mentioned in this study, the Australian coast emissions significantly contribute to the aerosol burden over the sea; I think some comparisons between the simulated and in-situ measured aerosol concentration overland could be helpful to evaluate the overall model performance. As shown in figure 4, the L11S simulation underestimates the BC compared to observation. It could be due to the sea breeze issue suggested by the authors, but it is also possibly caused by inappropriate emissions over the land (anthropogenic and natural sources). As shown in figure 3, the L11S and L11SCR simulations captured the DMSa signal well but relatively performed less well regarding SO4 concentration. It implies that the DMSa is not the only contributor to the SO4 over the ocean, and the overland emission could be an interesting one to investigate. Significantly, the background SO4 concentration over the sea could impact the sensitivity of aerosol burden to the coral reef emitted DMS.

We have now included additional figures in the Appendix, extending the model evaluation, including that for BC at AIRBOX. We note that BC at AIRBOX is also significantly underestimated, primarily due to some large peaks, which have previously been attributed to biomass burning. This does imply that the model emissions are not capturing the terrestrial/anthropogenic aerosol sources comprehensively. To investigate this further, multiple sensitivity tests around a range of terrestrial sources would be required, which is well outside our aims for this study. We have added text along these lines to the manuscript.

Line 252: 'We note that Appendix Figure C4 shows an evaluation of BC at AIRBOX, where modelled BC is also lower than that of the observations (by a factor of approximately 3.4). The majority of this underestimation is due to some very large peaks in the observations which Chen et al. (2018) attributed in part to biomass burning, but may also be a result of local vehicle movements. This may indicate that the model is not capturing some small-scale or transient terrestrial/anthropogenic emissions. This limitation is not detrimental to the results.'

2, More sensitivity tests are needed to evaluate the impact of the coral reef-derived emission on aerosols. Fiddes et al. (2021) suggested that the effect of coral reef-derived DMS depends on the back-ground aerosol loading. Therefore, it is interesting to test to what extent the reduction of the anthropogenic/natural (i.e., power plant, biomass burning, and sea salt) emissions could increase the impact of the coral reef-derived DMS? Adding these sensitivity tests enriches the importance of this study in the context of global energy and biomass burning trends.

While we agree that more sensitivity tests would be of great interest, further such simulations are not within our current capabilities. While we had made similar suggestions for future work in the conclusions already, (eg. performing simulations under pre- or post-industrial conditions), we have now also included further discussion around other sensitivity studies as suggested by the reviewer.

Line 459: 'Modelling studies that test the sensitivity of influence of coral-reef derived DMS to other aerosol burdens (eg. anthropogenic, biomass burning or sea-spray) would also be of significant value.'

**Minor comments**

**Line 23, missed a citation?**

Yes, a typo in the latex command. This has been fixed.

Line 26 to 29, the authors describe the DMS's in the radiative forcing, including direct and indirect

**effects. Could the authors give more detailed data two distinguish the two effects?**

The Fiddes et al. (2018) work quantified the overall radiative forcing of DMS, and while this included both indirect and direct forcing, the two were individually quantified, but where discussed on a region-by-region basis more qualitatively. We have added a small amount of text on this. We encourage the Reviewer to read the Fiddes et al. (2018) paper for more information.

Line ??: 'Fiddes et al. (2018) further found evidence of indirect effects predominantly occurring over stratocumulous decks over in the Southern Hemisphere.'

Line 30, what is the primary sources of DMS in the ocean?

DMS is primarily produced by phytoplankton in the ocean. We have made this more clear in the first line of the introduction. .

Line 15: 'Dimethyl sulfide (DMS) is an important precursor gas for aerosol formation. DMS is produced predominantly by marine organisms such as algae and phytoplankton.'

Line 229 to 231, do the authors check the simulated and observed BC concentration over land before concluding?

See response to major comment 1.

Line 266 to 267, please be more specified about the "internal model variability"

We have included some extra text indicating that some differences between the simulations is expected despite the nudging, that may not have been directly caused by the perturbations applied.

Line 302: 'We suggest that regions where the changes along the transects in the number and mass co-vary are likely due to internal model variability (eg. some variation between model simulations independent of perturbations, despite the nudging, is expected), rather than changes in the DMSa field.'

Line 350, does simulation become better when using hourly nudging? There should be some reanalysis meteorology data available with higher temporal resolution.

Overall we are quite happy with the performance of the nudging, as shown by the now-included meteorological evaluation plots in the appendix. The applied nudging achieved its aims of constraining the synoptic meteorology and limiting feedbacks resulting from the perturbations. We do not expect that a higher temporal resolution for the nudging will have large effects on the overall outcome of this work.

Line 384, Could authors specify what size bins in WRF-CHEM refer to the Aitken mode here?

We refer to the first bin as the Aitken mode in this work, which is 3.90625 x  $10^{-2}$  to 7.8125 x  $10^{-2}$  µm in size (see Fast et al. 2006).

Line 470: 'In MOSAIC, growth to Aitken mode particles (the smallest bin size in this simulation) is simulated implicitly as newly nucleated particle sizes are smaller than the smallest simulated aerosol size in the model.'